# Adaptive Uncertainty Estimation via High-Dimensional Testing on Latent Representations

**Tsai Hor Chan**
Department of Statistics and Actuarial Science
The University of Hong Kong
hchanth@connect.hku.hk

**Kin Wai Lau**
TCL AI Lab
Hong Kong
stevenlau@tcl.com

**Jiajun Shen**
TCL AI Lab
Hong Kong
shenjiajun90@gmail.com

**Guosheng Yin**
Department of Mathematics
Imperial College London
guosheng.yin@imperial.ac.uk

**Lequan Yu**[*]
Department of Statistics and Actuarial Science
The University of Hong Kong
lqyu@hku.hk

## Abstract

Uncertainty estimation aims to evaluate the confidence of a trained deep neural network. However, existing uncertainty estimation approaches rely on low-dimensional distributional assumptions and thus suffer from the high dimensionality of latent features. Existing approaches tend to focus on uncertainty on discrete classification probabilities, which leads to poor generalizability to uncertainty estimation for other tasks. Moreover, most of the literature require seeing the out-of-distribution (OOD) data in the training for better estimation of uncertainty, which limits the uncertainty estimation performance in practice because the OOD data are typically unseen. To overcome these limitations, we propose a new framework using data-adaptive high-dimensional hypothesis testing for uncertainty estimation, which leverages the statistical properties of the feature representations. Our method directly operates on latent representations and thus does not require retraining the feature encoder under a modified objective. The test statistic relaxes the feature distribution assumptions to high dimensionality, and it is more discriminative to uncertainties in the latent representations. We demonstrate that encoding features with Bayesian neural networks can enhance testing performance and lead to more accurate uncertainty estimation. We further introduce a family-wise testing procedure to determine the optimal threshold of OOD detection, which minimizes the false discovery rate (FDR). Extensive experiments validate the satisfactory performance of our framework on uncertainty estimation and task-specific prediction over a variety of competitors. The experiments on the OOD detection task also show satisfactory performance of our method when the OOD data are unseen in the training. Codes are available at `https://github.com/HKU-MedAI/bnn_uncertainty`.

---

[*]Corresponding Author

37th Conference on Neural Information Processing Systems (NeurIPS 2023).

# 1 Introduction

Deep neural networks (DNNs) have demonstrated state-of-the-art (SOTA) performances on many problem domains, such as computer vision [30, 31], medical diagnosis [24, 16, 38, 2], and recommendation systems [1, 3]. Despite their successes, most of the existing DNN designs can only recognize in-distribution samples (i.e., samples from training distributions) but cannot measure the confidence level (i.e., the uncertainties) in the prediction, especially for out-of-distribution (OOD) samples. Estimation of prediction uncertainties plays an important role in many machine learning applications [25, 37]. For instance, uncertainty estimation can help to detect OOD samples in the data and inspect anomalies in the system. Further, the uncertainty estimates can indicate the distributional shifts in the environment and thus facilitate learning in a non-stationary environment.

In response to the high demand, several attempts have been made to obtain a better estimate of the uncertainty, which can be roughly divided into two categories — Bayesian and non-Bayesian methods. Bayesian methods [4, 19] mainly operate with a Bayesian neural network (BNN), which introduces a probability distribution (e.g., multivariate Gaussian) to the neural network weights. This enables the model to address model-wise (i.e., epistemic) uncertainty by drawing samples from the posterior distribution. Non-Bayesian methods [32, 27, 10], on the other hand, assume a distribution on the model outputs (e.g., classification probabilities). The uncertainty scores can be obtained by evaluating a pre-determined metric (e.g., classification entropy) on the derived distribution.

However, most of the aforementioned methods are subject to several drawbacks: (1) They rely on strong assumptions (e.g., parametric models) on low-dimensional features and thus suffer from the curse of high dimensionality, leading to poor performance when the dimension of the output is high. (2) They are limited to classification problems, and existing methods explicitly make assumptions about the classification probabilities. This leads to poor generalizability in uncertainty estimation for other tasks, such as regression and representation learning. (3) Their performances heavily rely upon the feature encoder, which can be compromised when the features are of poor quality (e.g., the number of samples used to train the encoder is small). (4) They require a modification of the training loss, and thus additional training is needed when applying the methods to a new problem. Pretrained features cannot be directly applied to the OOD tasks without retraining for their proposed loss.

Observing the above limitations, we propose a new framework for uncertainty estimation. Our framework comprises two key components: one Bayesian deep learning encoding module and one uncertainty estimation module using the adaptable regularized Hotelling $T^2$ (ARHT) [26]. Our contributions are summarized as: (1) We formulate uncertainty estimation as a multiple high-dimensional hypothesis testing problem, and adopt a Bayesian deep learning module to better address the aleatoric and epistemic uncertainties when learning feature distributions. (2) We propose using the ARHT test statistic for measuring uncertainty and demonstrate the advantages of ARHT over the existing uncertainty measures, including its consistency and robustness properties. This enables us to design a data-adaptive detection method so that each individual data point can be assigned an optimal hyperparameter. Because it relaxes the strong assumptions on latent feature distributions, ARHT less sensitive to the feature quality. As result, our method can be interpreted as a post-hoc method as it works on feature distributions generated by any encoder (e.g., a pre-trained encoder). (3) We adopt the family-wise testing procedure to determine the optimal threshold of OOD detection, which minimizes the false discovery rate (FDR). (4) We perform extensive experiments on standard and medical image datasets to validate our method on OOD detection and image classification tasks compared to SOTA methods. Our proposed method does not require prior knowledge of the OOD data, while it yields satisfactory performance even when the training samples are limited.

# 2 Preliminaries

**Deep Neural Network**: A deep neural network (DNN) with $L$ layers can be defined as

$$f_l(\boldsymbol{x}; W_l, b_l) = \frac{1}{\sqrt{D_{l-1}}} \Big( W_l \phi(f_{l-1}(\boldsymbol{x}; W_{l-1}, b_{l-1})) \Big) + b_l, \quad l \in \{1, \dots, L\},$$

where $\phi$ is a nonlinearity activation function, $\boldsymbol{x}$ is the input, $D_{l-1}$ is the dimension of the input, $b_l \in R^{D_l}$ is a vector of bias parameters for layer $l$, and $W_l \in R^{D_l \times D_{l-1}}$ is the matrix of weight parameters. Let $\boldsymbol{w}_l = \{W_l, b_l\}$ denote the weight and bias parameters of layer $l$, and the entire trainable network parameters are denoted as $\boldsymbol{\theta} = \{\boldsymbol{w}_l\}_{l=1}^{L}$.

**Bayesian neural network (BNN)**: A BNN specifies a prior $\pi(\boldsymbol{\theta})$ on the trainable parameters $\boldsymbol{\theta}$. Given the dataset $\mathcal{D} = \{\boldsymbol{x}_i, y_i\}_{i=1}^N$ of $N$ pairs of observations and responses, we aim to estimate the posterior distribution of $\boldsymbol{\theta}$, $p(\boldsymbol{\theta}|\mathcal{D}) = \pi(\boldsymbol{\theta}) \prod_{i=1}^N p(y_i|f(\boldsymbol{x}_i; \boldsymbol{\theta}))/p(\mathcal{D})$, where $p(y_i|f(\boldsymbol{x}_i; \boldsymbol{\theta}))$ is the likelihood function and $p(\mathcal{D})$ is the normalization term.

**Pooled Sample Covariance Matrix:** Let $\{\boldsymbol{X}_{1j}\}_{j=1}^{n_1}$ be the $p$-dimensional embeddings from the training images and $\{\boldsymbol{X}_{2j}\}_{j=1}^{n_2}$ be the $p$-dimensional embeddings from the testing images. The pooled sample covariance is defined as

$$\boldsymbol{S}_n = \frac{1}{n-2} \sum_{k=1}^2 \sum_{j=1}^{n_k} (\boldsymbol{X}_{kj} - \bar{\boldsymbol{X}}_k)(\boldsymbol{X}_{kj} - \bar{\boldsymbol{X}}_k)^\top, \tag{1}$$

where $n = n_1 + n_2$, $n_k$ is the sample size and $\bar{\boldsymbol{X}}_k$ is the sample mean of the $k$-th set ($k = 1, 2$).

**High-Dimensional Test Statistics:** The Hotelling $T^2$ test statistic [15] is given by

$$T = \frac{n(n-p)}{p(n-1)} (\bar{\boldsymbol{X}}_1 - \bar{\boldsymbol{X}}_2)^\top \boldsymbol{S}_n^{-1} (\bar{\boldsymbol{X}}_1 - \bar{\boldsymbol{X}}_2). \tag{2}$$

The Hotelling $T^2$ test assumes $T \sim F(p, n-p)$ under the null hypothesis. To resolve the potential singularity of the covariance matrix, the regularized Hotelling $T^2$ (RHT) test statistic [5] loads an identity matrix $\boldsymbol{I}_p$ to $T$,

$$\mathrm{RHT}(\lambda) = \frac{n_1 n_2}{n_1 + n_2} (\bar{\boldsymbol{X}}_1 - \bar{\boldsymbol{X}}_2)^\top (\boldsymbol{S}_n + \lambda \boldsymbol{I}_p)^{-1} (\bar{\boldsymbol{X}}_1 - \bar{\boldsymbol{X}}_2), \tag{3}$$

where $\lambda$ is a tuning parameter. The ARHT test statistic, which will be formally introduced by Eq. 4 in Section 3.3, standardizes the RHT and addresses the skewness of Hotelling's $T^2$ when the feature dimension is high.

## 3 Methodology

Our uncertainty estimation framework comprises a Bayesian neural network encoder and an OOD testing module with ARHT as the uncertainty measure. Figure 1 provides an overview of our proposed framework, with the detailed algorithm given in the appendix. Ablation studies on key components of our framework are provided in Section 5.

### 3.1 Problem Definition and Test Hypothesis

Suppose that we have a set of samples $\{I_1, \ldots, I_n\}$ as the training set, and a set of testing samples $I'_1, \ldots, I'_n\}$ containing both in-distribution and OOD data. We aim to develop an uncertainty estimation framework that can accurately classify the test samples as either in-distribution or OOD, without seeing the OOD data during the training. Moreover, the framework can still perform well on its predictive task (e.g., classification) without sacrifice in uncertainty estimation. We set the null hypothesis $H_0 : \boldsymbol{\mu}_1 = \boldsymbol{\mu}_2$ and the alternative hypothesis as $H_1 : \boldsymbol{\mu}_1 \neq \boldsymbol{\mu}_2$, where $\boldsymbol{\mu}_1$ and $\boldsymbol{\mu}_2$ are the mean representations of the training and the testing samples, respectively.

### 3.2 Bayesian Neural Network as Encoder

We adopt a BNN encoder and train it with stochastic variational inference (SVI) [29] to learn the representation of the input. In SVI, a posterior distribution $p(\boldsymbol{\theta}|\boldsymbol{y})$ is approximated by a distribution $q$ selected from a candidate set $\mathcal{Q}$ by maximizing an evidence lower bound (ELBO): $\max_{q \in \mathcal{Q}} \mathbb{E}_{\boldsymbol{\theta} \sim q}[\log p(\boldsymbol{y}|\boldsymbol{\theta})] - \mathrm{KL}(q\|\pi)$, where $\mathrm{KL}(q\|\pi)$ is the Kullback–Leibler divergence between the variational posterior distribution $q$ and the prior distribution $\pi$, $p(\boldsymbol{y}|\boldsymbol{\theta})$ is the likelihood, and $\mathbb{E}_{\boldsymbol{\theta} \sim q}[\log p(\boldsymbol{y}|\boldsymbol{\theta})]$ represents the learning objective. This can be supervised (e.g., cross-entropy loss) or unsupervised (e.g., contrastive learning loss) learning. The KL divergence of two multi-variate Gaussian distributions is provided in the appendix. We use gradient descent to optimize the ELBO, i.e., to minimize the distance between prior and variational posterior distributions. We obtain the trained BNN encoder once the optimization step is completed. Since BNNs operate on an ensemble

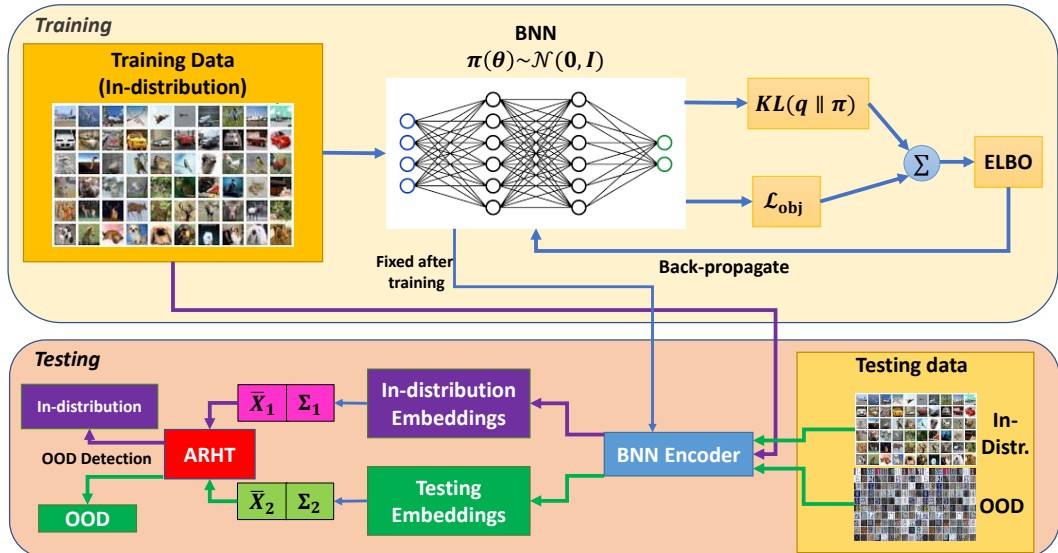

Figure 1: Workflow of our proposed uncertainty estimation framework. Our framework contains a BNN encoding module and an uncertainty estimation module using high-dimensional testing. The training objective $\mathcal{L}_{\text{obj}}$ can be either supervised (e.g., cross-entropy) or unsupervised (e.g., contrastive learning). The ELBO represents the evidence lower bound used for training BNN.

of posterior model weights, they can capture the epistemic uncertainties in the posterior predictive distribution $p(\boldsymbol{y}|\mathcal{D})$. Hence, using BNNs instead of frequentist architectures can better approximate the posterior distribution of the feature embeddings.

To obtain the uncertainty scores, we need to compute the sample means $\bar{\boldsymbol{X}}_1, \bar{\boldsymbol{X}}_2$ and covariance matrices $\boldsymbol{\Sigma}_1, \boldsymbol{\Sigma}_2$ for the training data (containing in-distribution data only) and the testing data (containing both in-distribution and OOD data). First, we sample $s$ weights for every data point in the training data, and generate representations for each data point using the sampled weights. In total, $n_1$ representations are generated from the training data. We can compute the mean $\bar{\boldsymbol{X}}_1$ and covariance matrix $\boldsymbol{\Sigma}_1$ of the training data using these $n_1$ representations. For each testing data point $x_t$, we sample $n_2$ weights from the variational posterior distribution, and generate $n_2$ representations $\{\hat{x}_{tj} = f(x_t; \boldsymbol{\theta}_j)\}_{j=1}^{n_2}$, where $\boldsymbol{\theta}_j$ is the $j$-th weight sample drawn from the trained posterior distribution $p(\boldsymbol{\theta}|\mathcal{D})$. We obtain the mean $\bar{\boldsymbol{X}}_2 = \sum_{j=1}^{n_2} \hat{x}_{tj}/n_2$ and the covariance matrix $\boldsymbol{\Sigma}_2 = \sum_{j=1}^{n_2}(\hat{x}_{tj} - \bar{\boldsymbol{X}}_2)(\hat{x}_{tj} - \bar{\boldsymbol{X}}_2)^\top/n_2$. Hence, we can compute the pooled covariance matrix $\boldsymbol{S}_n$ by Eq. (1).

### 3.3 High-Dimensional Testing as Uncertainty Measure

With the pooled sample covariance matrix, we can compute the ARHT test statistics by Eq. (3) and Eq. (4), where $\bar{\boldsymbol{X}}_1$ is the mean of training samples and $\bar{\boldsymbol{X}}_2$ is the mean of $n_2$ embeddings of each testing sample. To solve the skew $F$ distribution of RHT when $n \gg p$ for large $n$ and $p$ (as shown in Figure 2), we adopt the ARHT statistic to perform a two-sample test which has robust regularization of the covariance matrix [26]. In particular, the ARHT is given by

$$\text{ARHT}(\lambda) = \sqrt{p} \frac{p^{-1}\text{RHT}(\lambda) - \hat{\Theta}_1(\lambda, \gamma)}{\{2\hat{\Theta}_2(\lambda, \gamma)\}^{\frac{1}{2}}}, \tag{4}$$

where $\hat{\Theta}_1(\lambda, \gamma) = \{1 - \lambda m_F(-\lambda)\}/\{1 - \gamma(1 - \lambda m_F(-\lambda))\}$,

$$\hat{\Theta}_2(\lambda, \gamma) = \frac{1 - \lambda m_F(-\lambda)}{[1 - \gamma(1 - \lambda m_F(-\lambda))]^3} - \lambda \frac{m_F(-\lambda) - \lambda m_F'(-\lambda)}{[1 - \gamma(1 - \lambda m_F(-\lambda))]^4},$$

$$m_F(z) = \frac{1}{p}\text{tr}\{\boldsymbol{R}_n(z)\}, \quad m_F'(z) = \frac{1}{p}\text{tr}\{\boldsymbol{R}_n^2(z)\}, \quad \boldsymbol{R}_n(z) = (\boldsymbol{S}_n - z\boldsymbol{I}_p)^{-1}, \quad \gamma = \frac{p}{n}.$$

As a result, we have $\text{ARHT}(\lambda) \sim \mathcal{N}(0, 1)$ [26], which prevents the catastrophic skewness of $F$-distribution in high dimensions (see Figure 2) and yields smoother uncertainty scores. The ARHT can

Table 1: The OOD detection performance (in %) of our method, BNN-ARHT, compared to various competitors, using the LeNet [22] architecture. Standard deviations are given in brackets.

| | | OOD Datasets | | | | | |
| | | Fashion–MNIST | | OMNIGLOT | | SVHN | |
| Model | In-Distrib. | AUC | AUPR | AUC | AUPR | AUC | AUPR |
|---|---|---|---|---|---|---|---|
| MC Dropout [10] | MNIST | 99.33 (0.3) | 99.27 (0.3) | 99.85 (0.09) | 99.88 (0.07) | 99.96 (0.007) | **99.96 (0.002)** |
| Deep Ensembles [21] | MNIST | 90.70 (8.4) | 91.08 (7.7) | 99.70 (8.4) | 91.08 (7.7) | 99.21 (0.9) | 99.68 (0.4) |
| Kendall and Gal [19] | MNIST | 92.54 (2.6) | 92.77 (1.9) | 94.11 (4.9) | 93.40 (5.7) | 99.60 (0.5) | 99.13 (1.0) |
| EDL [32] | MNIST | 73.43 (16.0) | 80.22 (11.1) | 72.61 (8.6) | 81.42 (7.0) | 63.43 (1.2) | 85.09 (3.4) |
| DPN [27] | MNIST | 99.41 (0.2) | 99.37 (0.3) | 99.96 (0.03) | 99.96 (0.03) | 99.96 (0.01) | **99.96 (0.003)** |
| PostNet [4] | MNIST | 98.59 (0.4) | 94.70 (0.5) | — | — | — | — |
| Detectron [12] | MNIST | 75.57 (15.2) | 83.75 (29.9) | 95.71 (11.0) | 85.00 (30.0) | 77.92 (12.4) | 83.75 (37.3) |
| BNN-ARHT (Ours) | MNIST | **99.51 (0.4)** | **99.47 (0.3)** | **99.98 (0.01)** | **99.98 (0.004)** | **99.97 (0.007)** | **99.96 (0.004)** |
| MC Dropout [10] | CIFAR10 | 76.23 (5.6) | 74.21 (5.3) | 77.15 (2.2) | 79.0 (1.9) | 78.09 (1.2) | 84.35 (1.1) |
| Deep Ensembles [21] | CIFAR10 | 71.25 (3.0) | 75.32 (2.3) | 86.77 (3.6) | 90.35 (3.1) | 76.15 (5.3) | 82.62 (13.1) |
| Kendall and Gal [19] | CIFAR10 | 77.41 (17.3) | 77.00 (18.2) | 89.08 (17.8) | 90.93 (14.8) | 67.40 (3.1) | 71.44 (10.1) |
| EDL [32] | CIFAR10 | 67.81 (12.1) | 71.81 (11.5) | 77.53 (14.4) | 80.50 (11.7) | 69.57 (4.7) | 83.74 (3.4) |
| DPN [27] | CIFAR10 | 57.54 (1.7) | 68.29 (3.4) | 62.34 (3.7) | 70.49 (8.0) | 57.48 (4.4) | 77.76 (6.2) |
| PostNet [4] | CIFAR10 | — | — | — | — | 76.04 (1.6) | 69.30 (1.7) |
| Detectron [12] | CIFAR10 | 76.46 (15.3) | 71.63 (21.5) | 76.99 (15.9) | 91.00 (24.4) | 76.01 (13.6) | 90.00 (22.9) |
| BNN-ARHT (Ours) | CIFAR10 | **77.78 (5.0)** | **79.06 (6.8)** | **92.77 (1.8)** | **93.74 (1.0)** | **82.01 (1.2)** | **91.61 (0.3)** |

be interpreted as a more robust distance measure on embedding distributions compared to existing metrics such as Mahalanobis distance.

We select $\lambda$ from a predefined set using a data-adaptive method [26]. The set for grid search is chosen as $\{\lambda_0, 5\lambda_0, 10\lambda_0\}$ given the hyperparameter $\lambda_0$. For a testing data point $x_t$, we compute $Q(\lambda, \gamma; \boldsymbol{\xi}) = \sum_{k=0}^{2} \xi_k \hat{\rho}_k(-\lambda, \gamma)/\{\gamma \hat{\Theta}_2(\lambda, \gamma)\}^{1/2}$, for each $\lambda$ in the candidate set, where $\boldsymbol{\xi} = (\xi_0, \xi_1, \xi_2) \in \mathbb{R}^3$ is a pre-specified weight vector set as $\boldsymbol{\xi} = (0, 1, 0)$, $\hat{\rho}_0(-\lambda, \gamma) = m_F(-\lambda)$, $\hat{\rho}_1(-\lambda, \gamma) = \hat{\Theta}_1(\lambda, \gamma)$, and $\hat{\rho}_2(-\lambda, \gamma) = \{1 + \gamma \hat{\Theta}_1(\lambda, \gamma)\}\{p^{-1}\text{tr}\{\boldsymbol{S}_n\} - \lambda \hat{\rho}_1(-\lambda, \gamma)\}$. The optimal $\lambda$ is then selected by maximizing the $Q$ function, i.e., $\arg\max_\lambda Q(\lambda, \gamma; \boldsymbol{\xi})$. For each testing data point, we can proceed with the above selection process to determine the optimal $\lambda$ for the individual input. This makes the determination of uncertainty scores more flexible and data-adaptive.

### 3.4 Optimal Threshold Adjusting for Family-Wise Discovery Error

We obtain the area under the receiver operating characteristic curve (AUROC) and the area under the precision-recall curve (AUPR) using different thresholds. To determine the optimal threshold, we adopt a family-wise testing procedure to compute the $p$-values. It is necessary to balance the tradeoff between the power and the type I error rate of the test. We adopt the Benjamini–Hochberg (BH) procedure [35] procedure to adjust for multiple tests. The threshold of $p$-values is calibrated by the BH procedure, and we have a rejection set $\hat{R}$ of indices to the samples whose $p$-values are below the threshold, $\hat{R} = \{i \in \mathcal{I} : \hat{p}_i \leq \alpha\hat{k}/(mH_m)\}$, where $\hat{k} = \max\{k \in \mathcal{I} : \hat{p}_i \leq \alpha k(mH_m)\}$, $\mathcal{I} = \{1, \ldots, m\}$ is the set of indices corresponding to the $m$ tests, and $H_m = \sum_{j=1}^{m} 1/j$. The samples in the rejection set are then classified as the OOD sample. The BH procedure is shown to achieve a more powerful hypothesis testing performance [13].

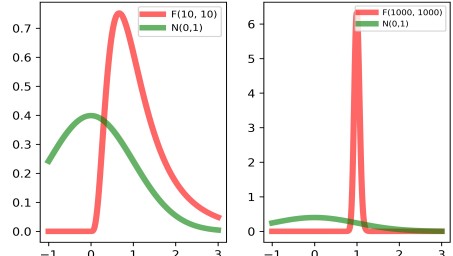

Figure 2: Comparison of the density functions of $\mathcal{N}(0, 1)$ and $F$-distribution under low (left) and high (right) dimensions.

## 4 Experiments

### 4.1 Datasets and Experiment Setting

We design OOD detection tasks on both standard and medical image datasets to demonstrate the application of our framework. We also evaluate our framework on the image classification

Table 2: The OOD detection performance (in %) on DRD [8], with the LeNet [22] architecture.

| Model | AUROC | AUPR |
|---|---|---|
| MC Dropout [10] | 59.52 (1.1) | 60.95 (6.0) |
| Kendall and Gal [19] | 91.06 (9.8) | 92.71 (7.8) |
| Deep Ensembles [21] | 59.67 (1.3) | 56.58 (2.3) |
| DPN [27] | 60.57 (1.1) | 65.32 (1.3) |
| EDL [32] | 53.01 (1.9) | 58.22 (9.6) |
| Detectron [12] | 90.74 (9.9) | 46.00 (31.4) |
| **BNN-ARHT (Ours)** | **93.44 (3.8)** | **95.44 (2.2)** |

Table 3: Classification performance (in %) on MNIST, with the LeNet [22] architecture.

| Model | Accuracy | F-1 Score |
|---|---|---|
| MC Dropout [10] | 98.48 | 98.89 |
| Kendall and Gal [19] | 98.78 | 98.65 |
| Deep Ensembles [21] | 97.90 | 97.89 |
| DPN [27] | 98.89 | 98.83 |
| EDL [32] | 21.24 | 12.88 |
| PostNet [4] | 99.12 | **99.06** |
| **BNN-ARHT (Ours)** | **99.26** | **99.06** |

task in comparison with the baseline methods to show that our framework does not sacrifice classification performance. For the image classification task, we benchmark the classification performance of the encoder trained on a holdout set of the in-distribution dataset (i.e., MNIST).

For the OOD detection task, we treat CIFAR 10 and MNIST as the in-distribution datasets, and Fashion–MNIST, OMNIGLOT, and SVHN as the OOD datasets. To validate the advantage of BNN encoding on limited observations, we compose a medical image benchmark using samples from the Diabetes Retinopathy Detection (DRD) dataset [8], with samples shown in Figure 3. We treat healthy

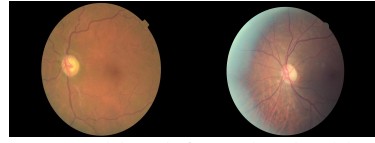

Figure 3: Healthy (left) and unhealthy (right) samples of the DRD dataset.

samples as in-distribution data and unhealthy samples as OOD data. Distinct from the settings in some existing works [27] which include OOD data in the training, we exclude the OOD data when training the feature encoder. We use the AUROC and AUPR as the evaluation metrics for OOD detection tasks, and adopt accuracy and the macro F1-score to evaluate the classification performance. Detailed definitions of the evaluation metrics and further descriptions of the datasets can be found in the appendix.

## 4.2 Competitive Methods

We compare our proposed framework, named as BNN-ARHT, with seven competitors — **(1) Deep ensembles** [21]: an ensemble of neural networks to address the epistemic uncertainties, and the number of models in the ensemble is set as 5; **(2) MC Dropout** [10]: a non-Bayesian method using dropout on trained weights to produce Monte Carlo weight samples [27]; **(3) Kendall and Gal [19]**: the first work addressing the aleatoric and epistemic uncertainties in deep learning; **(4) EDL [32]**: it estimates uncertainty by collecting evidence from outputs of neural network classifiers by assuming a Dirichlet distribution on the class probabilities; **(5) DPN [27]**: it assumes a prior network with Dirichlet distributions on the classification outputs; **(6) PostNet [4]**: it uses normalizing flow to predict an individual closed-form posterior distribution over predicted class probabilities; **(7) Detectron [12]**: it detects the change in distribution with the discordance between an ensemble of classifiers trained to agree on training data and disagree on testing data. Since Detectron operates on small samples from each dataset for OOD detection (e.g., 50 over 60,000 for CIFAR 10), we use a larger number of runs (i.e., 100) for a fair comparison. Five-fold cross-validation is applied to each of the competitive methods. We report the means and standard deviations over the runs for each metric.

## 4.3 Predictive and Uncertainty Estimation Performance

**Verify Uncertainty Quality by OOD Detection.** Tables 1 and 2 present the OOD detection results of our method compared with competitors on different pairs of datasets. We observe that existing methods, which assume having seen OOD data in the training, perform poorly in our settings, especially when the training observations are limited. Detectron [12] yields larger standard errors than other methods since it operates on small samples of the datasets. This validates the argument that existing methods heavily rely upon the availability of OOD data during the training. We also observe that our framework outperforms all the baselines on almost all OOD detection tasks, which demonstrates its satisfactory performance. In particular, our method shows a great improvement on the DRD dataset in which the number of samples is small, indicating the advantage of using a BNN encoder. As the MNIST dataset has dense feature representations, the OOD features can be easily distinguished. Hence, the OOD performance of the methods is relatively better on MNIST than on CIFAR10.

Table 4: The OOD detection performance (in %) of competitive methods under various model architectures [20, 22, 14]. CNN refers to the two-layer standard CNN architecture used by Malinin and Gales [27]. We use CIFAR10 as the in-distribution dataset and SVHN as the OOD dataset. 'Freq' represents the frequentist architecture and 'Bayes' represents the Bayesian architecture.

| | CNN | | LeNet | | AlexNet | | ResNet | |
|---|---|---|---|---|---|---|---|---|
| # of parameters (Freq) | 31,340 | | 62,006 | | 2,472,266 | | 11,699,522 | |
| # of parameters (Bayes) | 62,700 | | 125,112 | | 4,922,120 | | 12,372,904 | |
| Model | AUROC | AUPR | AUROC | AUPR | AUROC | AUPR | AUROC | AUPR |
| MC Dropout [10] | 63.85 | 74.79 | 68.58 | 78.53 | 74.72 | 82.57 | 61.46 | 76.04 |
| Deep Ensembles [21] | 78.44 | 89.10 | 61.01 | 62.69 | 72.14 | 59.83 | 58.11 | 78.26 |
| Kendall and Gal [19] | 67.77 | 80.87 | 55.36 | 69.82 | 70.88 | 73.15 | 50.01 | 83.86 |
| EDL [32] | 65.26 | 67.07 | 66.53 | 67.12 | 65.81 | 61.68 | 51.34 | 73.50 |
| DPN [27] | 63.98 | 77.30 | 57.44 | 73.84 | 67.10 | 80.75 | 76.36 | 86.46 |
| PostNet [4] | 73.68 | 66.85 | 76.04 | 69.30 | 81.67 | 76.52 | 82.19 | 79.43 |
| Detectron [12] | 73.84 | 88.50 | 76.01 | 90.00 | 80.58 | **85.00** | 78.27 | 89.50 |
| BNN-ARHT (Ours) | **85.36** | **89.37** | **82.01** | **91.61** | **82.10** | 70.04 | **88.16** | **93.19** |

**Predictive Performance Is Preserved.** As shown in Table 3, we also perform image classification to demonstrate the benefits of the post-hoc method (i.e., without modifying the original objective). We fix the neural network architecture as LeNet. For Kendall and Gal [19] and our method, we use the Bayesian counterpart of LeNet to perform the experiment. We observe that without modifying the original learning objective, the predictive performance of the encoder can be preserved and outperforms other methods. The BNN may underperform its frequentist counterpart due to the introduction of the KL regularization. Hence, in practice, a (pre-trained) frequentist encoder can be used to replace the BNN encoder if predictive performance is the focus. Section 5 shows the sacrifice in the OOD detection performance if a frequentist architecture is used.

**Visualization of Uncertainty Scores.** To better understand the uncertainty scores of in-distribution and OOD data under different uncertainty measures, Figure 4 presents the distributions of the ARHT uncertainty scores of in-distribution data (MNIST) and OOD data (Fashion-MNIST), respectively. We observe that the distributions of ARHT are of different shapes for in-distribution data and OOD data. This demonstrates the effectiveness of ARHT in identifying the unique characteristics of the distributions of the datasets.

## 5 Ablation Studies

**Different Neural Network Architectures.** We compare the performance with encoders under different architectures. We choose the standard CNN used in Malinin and Gales [27], LeNet [22], Alexnet [20], and ResNet18 [14] as the SOTA examples in encoding image features. For ResNet18, as the variance increases drastically with the increasing depth of the architecture, it is not feasible to replace all the layers with their Bayesian counterparts. Therefore, we replace only the last fully-connected layer and the second last convolutional layer with their Bayesian versions. Table 4 presents the comparison of OOD

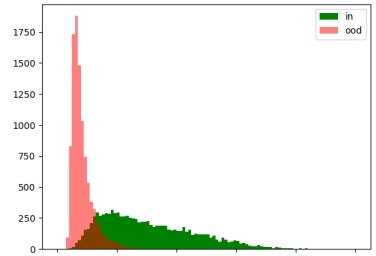

Figure 4: Distributions of the ARHT uncertainty scores for the in-distribution data (MNIST) and OOD data (Fashion-MNIST), respectively.

detection of our method with the competitors under different neural network architectures. We observe that our method is able to obtain satisfactory performance over the competitors when the neural network architecture changes.

**Effects of Key Hyperparameters.** We evaluate our method with a range of hyperparameters to assess their impacts on our method. Figure 5 presents the OOD detection performance with different values of $\lambda_0, n_2,$ and $p$. **(1) Initial loading value $\lambda_0$:** we observe that the performance of our model is robust to changes in $\lambda_0$. Since the ARHT relaxes the Gaussian assumption of embeddings, the change in the magnitude of the loading matrix would not heavily affect the covariance structure of the testing embeddings. Hence we can safely load $\lambda$ to the covariance matrix to resolve the singularity problem, with no concern about the decrease in performance; **(2) Number of training weight samples $n_1$:**

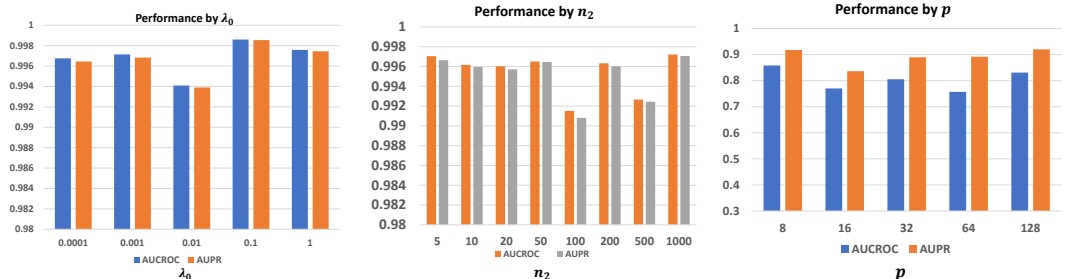

Figure 5: Performance of our method with different values of $\lambda_0$ (left), $n_2$ (middle), and $p$ (right) by fixing the network architecture as LeNet [22]. We use MNIST as the in-distribution dataset and Fashion-MNIST as the OOD dataset.

The size $n_1$ is controlled by the hyperparameter $s$ in our framework. We have conducted experiments with $s$ ranging from 1 to 5 (Figure 6). The pattern shows that the performance decreases when $s$ increases (i.e., more embedding samples from the in-distribution dataset). This demonstrates that the covariance structure affects ARHT more as $s$ increases, and the contribution of testing embeddings is less weighed, which leads to slightly decreasing performance. **(3) Number of testing weight samples** $n_2$: The number of testing weight samples $n_2$ is a key to approximate the testing embedding distributions. Since the approximation of the testing distribution is crucial to the performance of our method, we tune the hyper-parameter $n_2$ to determine the optimal number of testing embeddings to choose for each task. We observe that even if the number of testing samples is small (e.g., 5), ARHT can still capture the distributional difference between the in-distribution and OOD data. This leads to a consistent performance as $n_2$ varies. **(4) Embedding dimension** $p$: we also evaluate the OOD detection performance of our method with respect to the change in embedding dimensions ($p = 8, 16, 32, 64, 128$).

We observe that our method is robust to changes in feature dimensions. This enables our framework to perform OOD detection on the representation level with customized embedding dimensions, relaxing the constraint on classification problems.

**Feature Learning Objectives.** One key component of our framework is the BNN encoder, which is trained by supervision (i.e., cross-entropy loss for image classification) on standard datasets. We evaluate the robustness of our method when the encoder is trained with different objectives (i.e., unsupervised contrastive learning loss). We select the margin of

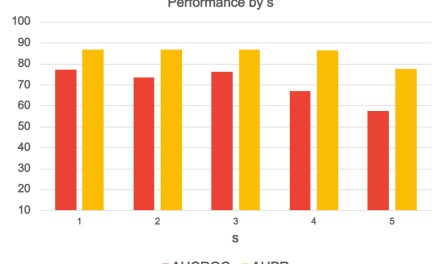

Figure 6: Ablation study with respect to $s$ (In-distribution: CIFAR 10, OOD: SVHN).

contrastive learning loss as 0.2. The performance in AUROC on the OOD detection task decreases slightly from 99.98 to 98.42 (for MNIST vs. OMNIGLOT OOD detection). This shows that our framework is sensitive to the quality of the encoder, and a supervised learning objective is preferred to improve the encoding performance.

## 6  Discussion: Impacts and Limitations of BNN and ARHT

We discuss why a BNN encoder is preferred for our framework, with Figure 7 illustrating the difference between features generated by BNN and frequentist DNN. A frequentist DNN gives one feature embedding to every input data point, and the uncertainty estimate ignores the covariance structure of the distribution because only the point estimate is provided. However, a BNN estimates the posterior embedding distribution for every data point, and the covariance structure can be incorporated to obtain a more accurate uncertainty estimate. Although some recent non-Bayesian method [32, 27] places parametric distributions on posterior embeddings (e.g., Dirichlet distribution on class probabilities), these parametric distributions are less accurate in approximating the posterior distributions than BNNs because the strong parametric assumption limits their capabilities in searching the candidate distributions.

To demonstrate why ARHT is preferred as an uncertainty metric, we further evaluate its performance over an array of uncertainty measures, including the maximum probability, entropy, and

differential entropy proposed by Malinin and Gales [27]. Detailed definitions of these metrics are presented in the appendix. We also include the frequently used Mahalanobis distance which possesses stronger assumptions on the Gaussianity of the embeddings. We fix the model architecture to be LeNet and compare its Bayesian and frequentist designs. Table 5 presents the summary of the comparison. We observe that using ARHT as the uncertainty score can achieve the best OOD detection performance than existing uncertainty measures when a BNN encoder is used.

Our method achieves a better performance using features generated by BNNs than those generated by frequentist counterparts, which demonstrates the advantage of using Bayesian encoders in our framework. However, BNN encoders require drawing $n_2$ samples of weights for posterior inference, which requires higher time complexity than the frequentist counterparts. Further, BNNs are limited by the scalability constraints [11], which makes them difficult to have deeper structures. How to make BNNs deep remains a challenging topic [11, 36, 7, 29].

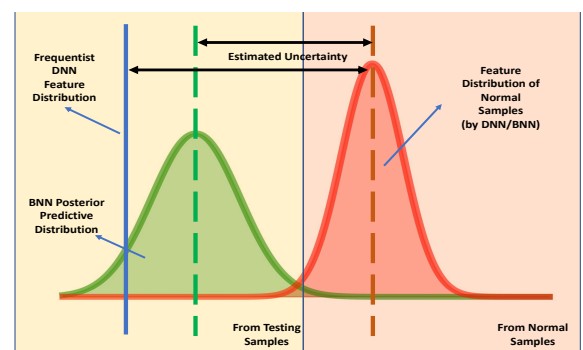

Figure 7: Comparison of features generated by a frequentist DNN and those by a BNN. The uncertainty estimated by a frequentist DNN ignores the covariance structure of the posterior distribution, where a BNN provides a distributional estimate for each testing sample.

**Support of Single-Sample Uncertainty.** As the training set (as the in-distribution set) is available (at least for training or fine-tuning the encoder) in most of the problems, one can use samples from training sets and the testing samples to compute ARHT. One exception is the zero-shot case where we only have the pre-trained encoder but no original data (i.e., in-distribution samples). In this case, most of the uncertainty estimation methods cannot work since they require at least in-distribution data to fit their parametric assumptions (e.g., concentration rates of the Dirichlet distributions in classification problems). However, one may still obtain ARHT as an uncertainty estimate using methods to reconstruct/generate pseudo-training data from the pre-trained models, which is not the focus of our work but warrants future research.

## 7 Related Works

**Uncertainty Estimation.** Estimating uncertainty in neural networks has become an increasingly important topic in deep learning. Existing uncertainty estimation methods can be divided into two classes: Bayesian methods [19, 23] and non-Bayesian methods [32, 27, 21, 4, 9, 10]. Bayesian methods address the model-wise (i.e., epistemic) uncertainties by learning a posterior distribution of weights. The uncertainties in neural networks can be approximated by the predictions given by the weights sampled from the posterior distributions.

Despite the success of these methods, most works only focus on classification uncertainties and rely upon arbitrary distributional assumptions on the class probabilities. For instance, DPN [27] assumes that the classification probabilities follow a Dirichlet distribution and train the OOD detector based on the KL divergence of the prior and posterior Dirichlet distributions. These methods are not generalizable to tasks other than classification. Furthermore, most existing methods [32, 27, 10] assume that the samples from the target domain are available when training the OOD detector, which is unrealistic in most applications.

Table 5: Performance of the ARHT against other uncertainty measures, with the LeNet [22] architecture, the in-distribution dataset CIFAR10, and the OOD dataset SVHN.

| Model | Frequentist | | Bayesian | |
|---|---|---|---|---|
| | **AUROC** | **AUPR** | **AUROC** | **AUPR** |
| Maximum Probability | 77.50 | 64.69 | 81.30 | 83.11 |
| Entropy | **80.22** | **88.47** | 82.49 | 78.14 |
| Differential Entropy | 79.59 | 86.98 | 66.15 | 75.27 |
| RHT | 79.06 | 62.82 | 82.29 | 64.07 |
| **ARHT** | 79.07 | 63.44 | **82.64** | **91.69** |

**Hypothesis Testing in High Dimension.** The task of uncertainty estimation can be redefined as a high-dimensional hypothesis testing problem. We report the detection of OOD samples if we reject the null hypothesis at significance level $\alpha$. The Hotelling $T^2$ test [15, 18] in Eq. (2) assumes $T \sim F(p, n - p)$ under the null hypothesis. The unnormalized version of $T$ in Eq. (2) is known

as Mahalanobis distance. However, the Hotelling test statistic suffers from poor robustness and consistency when $n$ and $p$ are large and even becomes undefined when $p > n$ or when $\boldsymbol{S}_n$ is singular [26]. This results in all test statistics (as uncertainty scores) concentrating on the singular point leading to trivial estimation of uncertainties. Figure 2 presents an example of comparisons of $F(10, 10)$ and $F(1000, 1000)$ to $\mathcal{N}(0, 1)$. Chen et al. [5] attempt to overcome this issue by proposing the RHT statistic, which resolves the singularity issue of the covariance matrix but yet the inconsistency and poor performance of Hotelling's $T^2$ statistic under the regime $p/n \to \gamma$. Li et al. [26] propose adaptable RHT and design an adaptive selection procedure for the loading parameter $\lambda$ based on the work of Chen et al. [5], which resolves the inconsistency of Hotelling's $T^2$.

**Bayesian Deep Learning.** SOTA DNN architectures [33, 14, 34] demonstrate significant success in tasks from different domains. Their designs enable them to mine high-dimensional features into low-dimensional representations and address the aleatoric uncertainties. Despite the success, the DNNs typically only yield maximum likelihood estimates of weights under the frequentist framework and cannot address epistemic uncertainties [19]. Existing works of BNN assign prior distributions to the neural network weights so as to obtain the posterior distributions of weights given the observed data [36, 11, 29, 17]. This enables BNN to provide a more accurate approximation to the target distribution and address epistemic uncertainties. Particularly when the training observations are limited, using a BNN can prevent overfitting and generate more representative feature distributions.

# 8 Conclusion

We propose a novel uncertainty estimation framework by introducing high-dimensional hypothesis testing to feature representations. We introduce the ARHT as the uncertainty measure which is adaptable to individual data point and robust compared to existing uncertainty measures. Our proposed uncertainty measure operates on latent features and hence can be generalized to any other tasks beyond image classification (e.g., regression or feature representation learning). Empirical evaluations on OOD detection and image classification tasks validate the satisfactory performance of our method over the SOTAs. Ablation studies on key components of the proposed framework validate the robustness and generalizability of our method to variations. One of the best potential applications of our framework is continual learning, where the proposed method can accurately measure the uncertainty as the domain shifts when encountering non-stationary environments. Our framework can be potentially applied to various settings where distributional shifts and OOD detection are vital, such as medical imaging, computational histopathology, and reinforcement learning.

**Acknowledgement.** We thank the anonymous reviewers, the area chair, and the program chair for their insightful comments on our manuscript. This work was partially supported by the Research Grants Council of Hong Kong (17308321), the Theme-based Research Scheme (T45-401/22-N), and the National Natural Science Fund (62201483).

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

**Overview.** In the appendix, we first provide more explanations on the ARHT test statistic in Section A. We then present a detailed summary of the datasets in Section B. Additional experiment results and analysis are reported in Section C Detailed descriptions of distributions used for BNN training are provided in Section D. We further present the implementation settings, hyperparameters and settings of baseline methods in Section E. We give the definitions of uncertainty measures used for comparison in Section F, together with an algorithm of our framework (Algorithm 1).

# A    Additional Discussion on ARHT

**Why using ARHT?** One of our motivations is to formulate uncertainty estimation as a hypothesis testing problem, where we interpret high-dimensional test statistics as distance measures. The ARHT can be viewed as a distance measure between each testing sample and the in-distribution sample, where a larger ARHT indicates the sample is more likely to be OOD. The detection of OOD is then determined by a threshold, while the optimal threshold can be obtained by the Benjamini–Hochberg (BH) procedure. One advantage of ARHT is that it directly operates on the sampling distributions of the latent features, and hence it does not require parametric assumptions on the latent features or logits (e.g., assuming a Dirichlet distribution on class probabilities), which makes it a more robust metric for uncertainty estimation.

**Motivation of loading $\lambda I_p$:** This is a standard way to ensure the covariance matrix to be positive definite (and thus invertible) and hence improve the numerical stability.

**Why does ARHT follow the standard normal distribution?** : The major part of Li et al. [26] is to prove why $\mathrm{ARHT}(\lambda)$ follows the standard Gaussian distribution. The proof is complex and hence is not focused in this paper. Intuitively, $\mathrm{ARHT}(\lambda)$ can be viewed as a "standardized" version of RHT (i.e., known as Mahalanobis distance) by its theoretical mean and SD, for which the derivations are detailed in Li et al. [26]

**Intuitive explanation and detailed derivation of the BH procedure:** Although the BH procedure may not be intuitive to the general audience, it is well-known in the statistics community due to multiple testing issues. Intuitively, we consider the OOD detection procedure for each testing image as a hypothesis testing problem. Then, such a procedure for the whole testing set can be viewed as a multiple-testing problem (i.e., conducting a large number of tests). However, applying a universal threshold for all tests (e.g., $\alpha = 0.05$) is too conservative and leads to many false discoveries. Hence, the BH procedure is applied to assign a threshold adaptable to each sample according to the $p$-values of all tests such that the false discovery rate (FDR) can be minimized.

# B    Additional Information on Datasets

Table 6 presents a summary of the datasets used for the experiments. We use eight image datasets to evaluate our method, including natural images and medical images.

Table 6: Summary of datasets, including the number of classes for classification, and the split of training and testing sets.

| Dataset | No. Classes | No. Training | No. Testing |
|---|---|---|---|
| **MNIST** | 10 | 60,000 | 10,000 |
| **Fashion-MNIST** | 10 | 60,000 | 10,000 |
| **OMNIGLOT** | 50 | 13,180 | 19,280 |
| **SVHN** | 10 | 73,257 | 26,032 |
| **CIFAR-10** | 10 | 60,000 | 10,000 |
| **CIFAR-100** | 100 | 60,000 | 10,000 |
| **TinyImageNet** | 200 | 80,000 | 20,000 |
| **DRD** | 2 | 50 | 100 |

**Diabetes Retinopathy Detection (DRD).** For this experiment, we define in-distribution samples as healthy (no DR; with label 0), and OOD samples as DR (mild, moderate, severe, or proliferative DR; corresponding to labels 1–4). We select 50 healthy images to train the encoder, and compute $\boldsymbol{\mu}_1$ and

$\Sigma_1$ from these samples using the trained encoder. For testing, we select 50 images as in-distribution data and 50 images as the OOD data. All images are resized to $64 \times 64$ for computational convenience. We train the encoder with a task to classify whether the input image is the left eye or right eye. The examples of healthy and DR are presented in Figure 8.

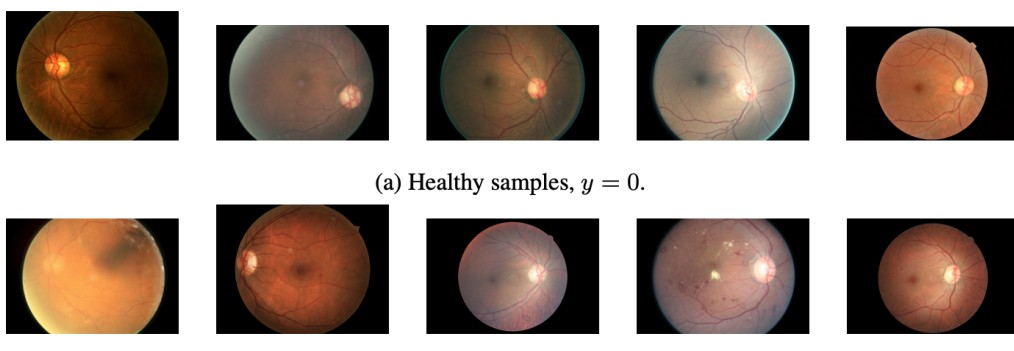

(a) Healthy samples, $y = 0$.

(b) Unhealthy samples, $y = 1$.

Figure 8: Health and unhealthy (DR) samples from the Diabetes Retinopathy Detection (DRD) dataset [8].

## C    Additional Experiment Results

We first present the OOD detection results on a more realistic dataset (i.e., TinyImageNet), with a more realistic architecture (e.g., ResNet50). We also include more baseline methods for comparison. Additionally, to study the effect of the encoder quality on OOD detection performance, we use the training accuracy at different epochs to measure the quality of the encoder, and assess how the OOD detection performance changes accordingly.

### C.1    More Realistic Datasets.

We have conducted additional experiments on OOD detection, with CIFAR10 as the in-distribution dataset and TinyImageNet as the OOD dataset. The results are presented in Tables 7 and 8, which show that the uncertainty estimation still performs satisfactorily when being generalized to larger datasets.

Table 7: The OOD detection performance (in %) of our method, BNN-ARHT, compared to various competitors, using the LeNet [22] architecture. We use CIFAR 10 as the in-distribution dataset and TinyImageNet as the OOD dataset.

| Model | AUC | AUPR |
|---|---|---|
| MC Dropout [10] | 66.98 | 64.46 |
| Deep Ensembles [21] | 66.41 | 63.97 |
| Kendall and Gal [19] | 63.23 | 63.06 |
| EDL [32] | 51.64 | 66.31 |
| DPN [27] | 61.68 | 58.33 |
| BNN-ARHT (Ours) | 67.77 | 66.74 |

### C.2    Scalability

We aim to use the Bayesian counterpart of a smaller architecture to demonstrate the capability of BNNs to generate latent feature distributions (for details see the discussion section in the main text). One can generate ideal feature distributions using very large frequentist vision models (e.g., ViT), which however induces enormous complexity in training and inference.

Table 8: The OOD detection performance (in %) of our method, BNN-ARHT, compared with various competitors, using the LeNet [22] architecture. We use TinyImageNet as the in-distribution dataset and CIFAR10 as the OOD dataset.

| Model | AUC | AUPR |
|---|---|---|
| MC Dropout [10] | 64.36 | 60.47 |
| Deep Ensembles [21] | 66.41 | 63.97 |
| Kendall and Gal [19] | 58.54 | 55.29 |
| EDL [32] | 50.37 | 70.39 |
| DPN [27] | 59.59 | 59.87 |
| BNN-ARHT (Ours) | **69.27** | **71.54** |

Table 9: The OOD detection performance (in %) of our method, BNN-ARHT, compared with various competitors, using the ResNet50 architecture. We use CIFAR 10 as the in-distribution dataset and SVHN as the OOD dataset.

| Model | AUC | AUPR |
|---|---|---|
| MC Dropout [10] | 68.32 | 78.24 |
| Deep Ensembles [21] | 65.13 | **82.19** |
| Kendall and Gal [19] | 72.24 | 81.43 |
| EDL [32] | 51.21 | 73.78 |
| DPN [27] | 62.33 | 79.11 |
| Detectron [12] | 73.16 | 82.5 |
| BNN-ARHT (Ours) | **73.46** | 78.27 |

A related ablation experiment comparing the frequentist and Bayesian architecture is presented in the main text. We additionally conduct an experiment with the Bayesian model architecture scaled up to ResNet50. We further conduct an experiment using the frequentist ResNet50 (the hypothesis test reduces to a one-sample test) and the testing AUROC is 72.77. These results validate the scalability of our method to large and modern vision architectures.

### C.3 Encoder Quality

To assess the effect of the encoder quality, we fix the encoder (e.g., LeNet) when comparing our framework with the current SOTA methods. Since the baseline methods are trained on classification problems (e.g., the CIFAR 10 image classification and DRD auxiliary task), we use the training accuracy at different epochs to measure the quality of the encoder. We observe that the OOD detection performance is monotonously improved with the increase in the training accuracy (i.e., the encoder quality). Figure 9 presents an example of the OOD detection experiment on CIFAR10 and TinyImageNet.

### C.4 Influence of $n_2$.

Figure 11 presents the OOD detection results under a more complex setting (In-distribution: CIFAR10, OOD: SVHN). We observe a similar pattern to that in Figure 5 in the main text. This shows that the sample covariance is more influenced by the $n_1$ training/in-distribution samples, making the test statistics reflect more the training distribution (hence the overall consistent pattern). Future work on variance-adjusted test statistics may put more weight on the feature distributions of testing samples, so that we would more easily observe performance improvement as $n_2$ increases in this case.

### C.5 Under the Regression Setting.

We choose OOD as the benchmark task since it is the most common benchmark for uncertainty estimation. We additionally construct a regression setting with two multivariate Gaussian distributions of different means and variances indicating different distributions. Most of the uncertainty estimation frameworks cannot be applied to this setting because they only work under the classification settings.

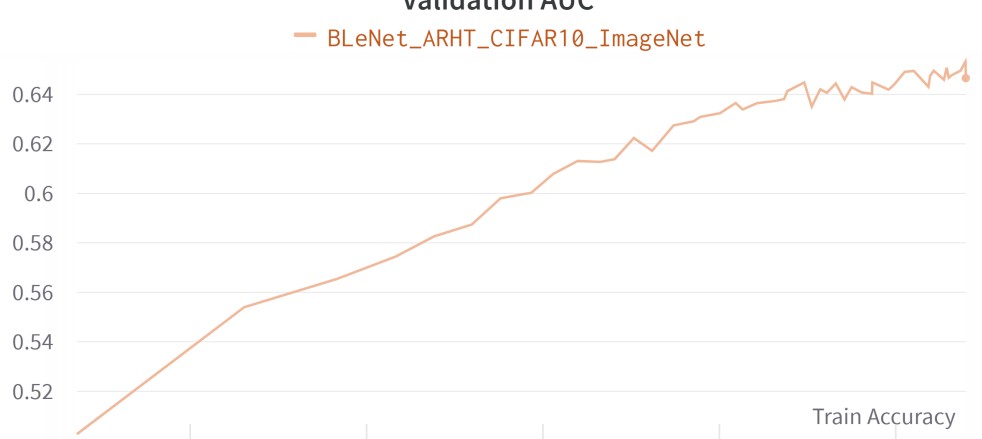

Figure 9: The OOD detection performance in terms of AUROC with respect to the training accuracy of the encoder. We adopt CIFAR 10 as the training/in-distribution dataset and TinyImageNet as the OOD data.

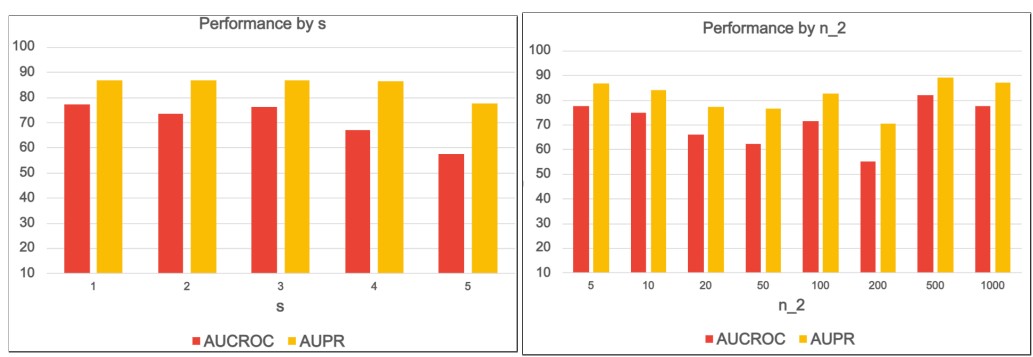

Figure 10: Ablation study with respect to $s$. We use CIFAR10 as the in-distribution dataset and SVHN as the OOD dataset.

Figure 11: Ablation study with respect to $n_2$. We use CIFAR10 as the in-distribution dataset and SVHN as the OOD dataset.

The results in Table 10 show that our method also achieves satisfactory performance under the regression settings, demonstrating its generalizability to other tasks.

### C.6   Comparison with More Baselines

More baselines on uncertainty estimation are added for comparison: (1) I-EDL [6]: use the Fisher information matrix to measure the informativeness of evidence carried by each sample; and (2) RKL-PN [28]: prior networks trained with the reverse KL divergence. Table 11 presents the additional results, from which we observe that our method still achieve the state-of-the-art performance with more baselines included.

## D   Multivariate Gaussian Distribution

The multivariate Gaussian distribution is crucial for the approximation of a vanilla BNN [17]. We provide the formal definition and its important properties in this section. The density of a multivariate

Table 10: The OOD detection performance (in %) of our method, BNN-ARHT, compared with various competitors, using the ResNet50 architecture. We constructed a simulated regression setting where $\mathcal{N}(\boldsymbol{\mu}, \boldsymbol{\Sigma})$ is the distribution for in-distribution data and $\mathcal{N}(-\boldsymbol{\mu}, \boldsymbol{\Sigma})$ is the distribution for OOD data. The auxiliary regression task is to predict the norm of the sampled vector using an MLP with two layers. We choose $\boldsymbol{\mu} = [0.5, \ldots, 0.5]^\top$, $\boldsymbol{\Sigma} = 9\boldsymbol{I}_p$, and $p = 128$.

| Model | AUROC | AUPR |
|-------|-------|------|
| MC Dropout [10] | 62.12 | 63.35 |
| Deep Ensembles [21] | 73.18 | 70.45 |
| Kendall and Gal [19] | 67.00 | 70.00 |
| BNN-ARHT (Ours) | **73.52** | **72.99** |

Table 11: The OOD detection performance (in %) of our method, BNN-ARHT, compared with various competitors, using the LeNet architecture. Standard deviations are given in brackets.

| | | OOD Datasets | | | | |
|---|---|---|---|---|---|---|
| | | **Fashion–MNIST** | | | **SVHN** | |
| **Model** | **In-Distrib.** | **AUC** | **AUPR** | **In-Distrib.** | **AUC** | **AUPR** |
| MC Dropout | MNIST | 99.33 (0.3) | 99.27 (0.3) | CIFAR10 | 78.09 (1.2) | 84.35 (1.1) |
| Deep Ensembles | MNIST | 90.70 (8.4) | 91.08 (7.7) | CIFAR10 | 76.15 (5.3) | 82.62 (13.1) |
| Kendall and Gal | MNIST | 92.54 (2.6) | 92.77 (1.9) | CIFAR10 | 67.40 (3.1) | 71.44 (10.1) |
| EDL | MNIST | 73.43 (16.0) | 80.22 (11.1) | CIFAR10 | 69.57 (4.7) | 83.74 (3.4) |
| DPN | MNIST | 99.41 (0.2) | 99.37 (0.3) | CIFAR10 | 57.48 (4.4) | 77.76 (6.2) |
| PostNet | MNIST | 98.59 (0.4) | 94.70 (0.5) | CIFAR10 | 76.04 (1.6) | 69.30 (1.7) |
| Detectron | MNIST | 75.57 (15.2) | 83.75 (29.9) | CIFAR10 | 76.01 (13.6) | 90.00 (22.9) |
| RKL-PN | MNIST | — | 78.45 (3.1) | CIFAR10 | 57.89 (1.8) | 61.41 (2.8) |
| I-EDL | MNIST | 98.49 (0.3) | 98.89 (0.3) | CIFAR10 | — | 83.26 (2.4) |
| BNN-ARHT (Ours) | MNIST | **99.51 (0.4)** | **99.47 (0.3)** | CIFAR10 | **82.01 (1.2)** | **91.61 (0.3)** |

Gaussian distribution is defined as

$$p(\boldsymbol{x}; \boldsymbol{\mu}, \boldsymbol{\Sigma}) = \frac{1}{(2\pi)^{\frac{n}{2}} |\boldsymbol{\Sigma}|^{\frac{1}{2}}} \exp\left\{ -\frac{1}{2}(\boldsymbol{x} - \boldsymbol{\mu})^\top \boldsymbol{\Sigma}^{-1}(\boldsymbol{x} - \boldsymbol{\mu}) \right\},$$

where $\boldsymbol{\mu} \in \mathbb{R}^p$ is a $p$-dimensional mean vector and $\boldsymbol{\Sigma} \in \mathbb{R}^{p \times p}$ is the covariance matrix. The KL divergence between two multivariate normal distributions $\mathcal{N}(\boldsymbol{\mu}_1, \boldsymbol{\Sigma}_1)$ and $\mathcal{N}(\boldsymbol{\mu}_2, \boldsymbol{\Sigma}_2)$ is given by

$$\mathrm{KL}(\mathcal{N}(\boldsymbol{\mu}_1, \boldsymbol{\Sigma}_1) \| \mathcal{N}(\boldsymbol{\mu}_2, \boldsymbol{\Sigma}_2)) = \frac{1}{2}\left[ \log\frac{|\boldsymbol{\Sigma}_2|}{|\boldsymbol{\Sigma}_1|} - p + \mathrm{tr}\{\boldsymbol{\Sigma}_2^{-1}\boldsymbol{\Sigma}_1\} + (\boldsymbol{\mu}_2 - \boldsymbol{\mu}_1)^\top \boldsymbol{\Sigma}_2^{-1}(\boldsymbol{\mu}_2 - \boldsymbol{\mu}_1) \right].$$

# E   Baseline Methods and Implementation Details

**Implementation Details.** We present additional implementation details and hyperparameter settings. We first provide the key settings and adaptations applied to the baseline methods for reproducibility. We follow the default settings for other fine-grained parameters (e.g., learning rates).

The proposed method is implemented in Python with *Pytorch* library on a server equipped with four NVIDIA TESLA V100 GPUs. The dropout ratio of each dropout layer is selected as 0.2. All models are trained with 100 epochs with possible early stopping. We use the *Adam* optimizer to optimize the model with a learning rate of $5 \times 10^{-5}$ and a weight decay of $1 \times 10^{-5}$. Data augmentations such as color jittering and random cropping and flipping are applied as a regularization measure.

**Hyperparameter Settings.** The hyperparameter settings for BNN training and ARHT testing are given as follows:

- Prior mean of weights — sampled from $\mathcal{N}(-3, 0.01)$
- $s = 5$

- $n_2 = 300$
- $\lambda_0 = 0.01$

**Additional Settings of Baseline Methods.** We further introduce the experimental settings of baseline methods:

- Deep ensembles [21]: Set the number of ensembles as 5.
- MCDropout [10]: Set the dropout ratio as 0.2 for both training and inference.
- Kendall and Gal: Set the number of inference weight samples as 20.
- Detectron [12]: Set the number of runs as 100.
- PostNet [4]: Because the original codes operate on the features extracted from the standard datasets, this method cannot be generalized to new datasets (e.g., SVHN) due to unavailability of the data processing codes.

# F  Uncertainty Measures

We describe the uncertainty measures used in the OOD misclassification task in this section. These definitions are well-known and summarized in Malinin and Gales [27],

- Entropy:

$$H[p(\boldsymbol{\mu}|\mathcal{D})] = -\sum_{c=1}^{K} p(w_c|\mathcal{D}) \ln p(w_c|\mathcal{D}),$$

   where $P(w_c|\mathcal{D})$ is the predictive probability of class $c$, and $K$ is the number of classes for classification.

- Maximum probability: we take the maximum predicted probability $\mathcal{P}$ from all classes as the confidence score,

$$\mathcal{P} = \max_c P(w_c|\mathcal{D}).$$

- Differential entropy:

$$I[y, \boldsymbol{\mu}|\mathcal{D}] = -\int_{S^{K-1}} p(\boldsymbol{\mu}|\mathcal{D}) \ln p(\boldsymbol{\mu}|\mathcal{D}) d\boldsymbol{\mu}$$

   where $S^{K-1}$ is the supporting set, and $\boldsymbol{\mu}$ is the predictive class probability assumed to follow a Dirichlet distribution.

- Accuracy: the fraction of correct predictions to the total number of ground truth labels.

- F-1 score: The F-1 score for each class is defined as

$$\text{F-1 score} = 2 \cdot \frac{\text{precision} \cdot \text{recall}}{\text{precision} + \text{recall}}$$

   where 'recall' is the fraction of correct predictions to the total number of ground truths in each class and 'precision' is the fraction of correct predictions to the total number of predictions in each class.

- AUROC: the area under the receiver operating curve (ROC) which is the plot of the true positive rate (TPR/Recall) against the false positive rate (FPR).

- AUPR: the area under the precision-recall curve. Note that the AUPR for binary classification is sensitive to the distribution of positive and negative classes. Hence, the higher AUPR does not necessarily imply a better model performance.

# G  Algorithm

Algorithm 1 gives the detailed workflow of our proposed uncertainty estimation framework.

**Algorithm 1** Our proposed uncertainty estimation framework
***

**Input:**
The prior distribution of weights of BNN encoder $\pi(\boldsymbol{\theta}) \sim \mathcal{N}(0, \boldsymbol{I})$;
Training data $\mathcal{D}_{tr} = \{\boldsymbol{x}_i, y_i\}_{i=1}^{N_{tr}}$;
Testing data $\mathcal{D}_{te} = \{\boldsymbol{x}_i, y_i\}_{i=1}^{N_{te}}$;
Hyperparameters $\mu_0, \rho_0, n_2$;
Initial variational posterior distribution $q(\boldsymbol{\theta}) \sim \mathcal{N}(\boldsymbol{\mu}, \log(1 + \exp(\boldsymbol{\rho})))$ with initial parameters
$\boldsymbol{\mu} = \mu_0 \mathbf{1}$ and $\boldsymbol{\rho} = \rho_0 \mathbf{1}$
**Output:** The uncertainty scores

1: **for** $(\boldsymbol{x}_i, y_i)$ in $\mathcal{D}_{tr}$ **do** ▷ Train BNN encoder
2:      Draw weight sample $\boldsymbol{\theta}$ from $q(\boldsymbol{\theta})$
3:      $\hat{y}_i = f_{\boldsymbol{\theta}}(\boldsymbol{x}_i)$ ▷ Forward propagation
4:      Compute task-specific loss $\mathcal{L}_{\mathrm{obj}}$
5:      Compute $\mathrm{KL}(q\|\pi)$ and hence the ELBO
6:      Backpropagate the ELBO to update $\boldsymbol{\mu}$ and $\boldsymbol{\rho}$
7: **end for**
8: Compute $\boldsymbol{\mu}_1 \in \mathbb{R}^p$, $\boldsymbol{\Sigma}_1 \in \mathbb{R}^{p \times p}$ ▷ Obtain summary statistics of training
9: **for** $(\boldsymbol{x}_i, y_i)$ in $\mathcal{D}_{tr}$ **do** ▷ OOD Detection
10:      Compute $\boldsymbol{\mu}_2 \in \mathbb{R}^p$, $\boldsymbol{\Sigma}_2 \in \mathbb{R}^{p \times p}$
11:      Compute the pooled sample covariance matrix by Eq. (1)
12:      Compute ARHT by Eq. (4) as the uncertainty score
13:      Detect OOD samples using the uncertainty score under family-wise adjusted threshold
14: **end for**
***

