# OpenReview forum: "Adaptive Uncertainty Estimation via High-Dimensional Testing on Latent Representations"
_NeurIPS.cc/2023/Conference — NeurIPS 2023 poster_

### Official Review · Reviewer_HkqB · 2023-06-26

**Soundness:** 4 excellent
**Presentation:** 3 good
**Contribution:** 3 good
**Rating:** 6
**Confidence:** 4

**Summary:**

The paper presents a new framework for uncertainty estimation in baysian neural networks. The core contribution is formulating uncertainty estimation as multiple high-dimensional hypothesis testing problem and deriving the test statistics necessary. The paper then present multiple empirical results, showing good performance in well established benchmarks and a excellent ablation study of relevant hyperparameters of their proposed method.

**Strengths:**

Originality:
The core idea of the paper of formulating uncertainty estimation as hypothesis testing seems novel. The paper essentially combines already established methods, BNNs with hypothesis testing (both which are well know), into a new exiting direction for this line of work.

Quality:
The paper is of high quality, both in outlining the method and their experimental section. For example, it is nice to see standard deviations added for nearly all results and a lot of relevant baselines to compare against. The inclusion of DRD datset, to test the method on a more "real-life" dataset is also a welcome addition and really shows the that the methods does not only work on benchmark datasets.

Clarity:
The paper is well written and easy to follow.

Significance:
The paper does open a new sub-direction within uncertainty estimation, regarding the use of hypothesis testing. Only time will tell how influential this will be. I am not truly convinced that the ARHT testing size is the correct way to go as it still involves an assumption of sub-Gaussuanity on its input.

The core strength of the paper is its section 5 regarding the ablation studies. The framework presented in the paper includes a lot of "moving parts" and therefore without a thoroughly investigation on which parts of this many layered method, it would be impossible to say which had a significant impact on the performance of the framework. The ablation study does a good job highlighting how each component feeds into the framework.

Also, thanks to the authors for providing anonymised code, helped understand parts of the paper.

**Weaknesses:**

In L56-58 the authors claims that "We formulate uncertainty estimation as a multiple high-dimensional hypothesis testing problem, and propose a Bayesian deep learning module to address better the aleatoric and epistemic uncertainties when learning feature distributions.". While I agree with the first part of the sentence, I would say the second part is not a contribution as too my knowledge the BNN framework used in the paper is not novel, which there is nothing wrong with, but should not be listed as a contribution.
In that regard I think the authors could make it more explicit that the first part of their framework is not unique in any sense, and therefore could be any BNN. Regarding results, similarly the authors should be more honest about their results. L206-206 state that their method is "superior", however if accounting for standard deviations the results only seems significant in a single case.
Thus overall, authors should adjust the claims in their paper to better reflect the work presented.

A small correction/question: From L130-139, the authors provide a bunch of equations, where the choice of parenthesises seems very arbitrarily. Sometimes {} is used, sometimes [] is used. sometimes () is used. It is confusing. Additionally in L139 there seems to be missing a "/" between what I assume is the nominator and denominator in a fraction.

**Questions:**

* Have the authors any thoughts on the influence of n_1, e.g. the size of the training set? For all datasets tested in the paper this is fixed, but the number of samples in the training distribution compared to the test distribution seems to be important for this method to work.

* Regarding Figure 5: Have the authors tried out this kind of ablation studies on other dataset combinations to see if they experience similar pattern? Mnist-FMnist is one of the easier cases for OOD and I wonder if this has something to do with the observed robustness of the method.

* Regarding Figure 5: The results from varying n_2 is strange to me. I would assume as we sample more test datapoints we get more certain about the test distribution making it easier to distinguish for the test distribution, however this does not seem to be the case. Does the authors have an explanation for this?

**Limitations:**

As the authors state in L300-302, BNNs are still hard to scale which seems to be the main limitation this line of work. That said, assuming that in future this gets solved this method seems very easy to apply.

Another limitation of the method, seems to be that it is limited to OOD, or at least tasks where distributions are compared. OOD is an important tasks, however just estimating the predictive uncertainty of a single sample is similarly important, which the framework does not seem to support.

---

> ### Author Rebuttal · Authors · 2023-08-09
>
> We greatly appreciate your positive comments on our manuscript and your insightful summary of the impacts and novelty of our work. We take great pleasure to respond to several intriguing discussions you raised as follows.
>
> >**Q1. Improving Clarity of Claims.**
>
> Thank you for your suggestions on improving the clarity of several critical claims in our manuscript. For the use of BNNs, we agree that our contribution is to adopt BNN to generate better latent distributions of samples (in contrast to conventional uncertainty estimation methods assuming parametric distributions), instead of proposing a new BNN learning framework. We will revise this claim and specify that our method adopts existing BNN advances instead of proposing a new BNN framework for better clarity. Also, we agree that the word "superior" may be inaccurate to describe the experiment results, and thus we will adjust the claims with a more careful choice of words (e.g., removing the "superior") in the future stage.
>
> >**Q2. Influence of $n_1$.**
>
> Thank you for pointing this out and we think this is an interesting discussion on future designs of the sample sizes $n_1$ and $n_2$. We think this is also related to your question on the strange performance varying by $n_2$. The size $n_1$ is controlled by the hyperparameter $s$ in our framework. We have conducted experiments with $s$ ranging from 1 to 5. Please see Figure 1 in the rebuttal file for the results. The pattern shows that the performance is decreasing when $s$ is increasing (i.e., more embedding samples from the in-distribution dataset). This demonstrates that covariance structure affects ARHT more as $s$ increases, and the contribution of testing embeddings is less weighed, which leads to slightly decreasing performance.
>
> >**Q3. Influence of $n_2$.**
>
> Figure 2 in the rebuttal file presents the OOD detection results under a more complex setting (CIFAR10 to SVHN). We can observe a similar pattern as the one shown in the manuscript. This shows that the sample covariance is more influenced by the $n_1$ training/in-distribution samples, making the test statistics reflect more the training distribution (hence the overall consistent pattern). Future work on variance-adjusted test statistics may put more weight on the feature distributions of testing samples, where we may more easily observe improving performance as $n_2$ increases in this case.
>
> >**Q4. Support of Single-Sample Uncertainty.**
>
> Since the training set (as the in-distribution set) is available (at least training or fine-tuning the encoder) for most of the problems, one can use samples from training sets and the testing samples to compute ARHT. One significant exception is the zero-shot case when we only have the pre-trained encoder without the original data (i.e., in-distribution sample). In this case, most of the uncertainty estimation methods may not work since they require at least in-distribution data to fit their parametric assumptions (e.g., concentration rates of the Dirichlet distributions in classification problems). However, one may still obtain ARHT as an uncertainty estimate using methods to reconstruct/generate pseudo-training data from the pretrained models, which is not the focus of our work but an interesting direction in the future.
>
> >**Q5. Use of Brackets.**
>
> Thank you for the suggestions. We will make the use of brackets consistent in the future stage given that it is not possible to update the manuscript at this stage. We have double-checked that the expression is correct for L139 (i.e., $\hat{\rho}_2(-\lambda, \gamma)$ is the product of two expressions $(1 + \gamma \hat{\Theta}_1(\lambda, \gamma))(p^{-1}{\rm tr} (\boldsymbol{S_n})  - \lambda \hat{\rho}_1(-\lambda, \gamma))$).

---

> > ### Comment · Reviewer_HkqB · 2023-08-15
> > **Answer to authors**
> >
> > I thank the authors for the response to my questions. I am glad that the authors are willing to change some of the language in their paper to better reflect the paper contributions. Additionally, I am happy with the response to my questions regarding some of the hyperparameters of the method. I hope some of this will be included in the camera-ready version or appendix in the future.
> >
> > Based on the response from the authors and the other reviews, I will be keeping my score for now.

---

> > > ### Author Response · Authors · 2023-08-17
> > >
> > > Dear reviewer HkqB,
> > >
> > > We are glad that you are happy with our response. Yes we will include the discussion and additional results in our future version. Please let us know if there are any further questions on our manuscript or response and we are pleased to answer. Thank you again for your precious time and effort on our manuscript.
> > >
> > > Best Regards,
> > >
> > > Authors

---

### Official Review · Reviewer_VpVz · 2023-06-30

**Soundness:** 3 good
**Presentation:** 2 fair
**Contribution:** 3 good
**Rating:** 6
**Confidence:** 4

**Summary:**

This paper proposes a framework to detect out-of-distribution (OOD) data via high-dimensional testing on latent representations.

The proposed framework consists of:
- a Bayesian Neural Network that, for any input, can produce an ensemble of latent presentations by sampling the posterior of the weights
- a hypothesis testing procedure that computes the adaptable regularized Hotelling's $T^2$ score (ARHT) as a measure of uncertainty, and classifies the input as OOD when the ARHT score is larger than a threshold calibrated by the Benjamini-Hochberg (BH) procedure

The paper demonstrates the superior performance of the proposed framework on standard OOD detection benchmarks and a medical image dataset.

**Strengths:**

- The idea of using Bayesian NN to generate an ensemble of latent representations, and then leveraging the ARHT score as a measure of uncertainty is novel, interesting, and useful for many practical applications

- The paper is generally well written and easy to follow.

**Weaknesses:**

- A major weakness of this paper is that, for readers who do not have a statistical background, the ARHT score, i.e., equations (1)-(4), appear to come from nowhere. Particularly, I think the paper will be much easier to appreciate if comments and explanation could be made about:
    - What is the intuitive explanation of the Hotelling's $T^2$ test statistic $T$? Why should we believe it is a good metric for separating in-distribution and OOD samples?
    - What is the motivation of loading a scaled identity matrix to $T$? In the related work section (which appears at the end of the paper) you seem to state that was due to the potential singularity of the matrix $S_n$. I believe you should state this earlier when you present the equation (2) and (3).
    - Why the ARHT score $T(\lambda)$ satisfies a Gaussian distribution $\mathcal{N}(0,1)$? This claim seems to come from reference [24]. I believe you need to briefly explain this to the reader.

- Section 3.4 describes the B-H procedure without intuitive explanation and detailed derivation. I think more contents are need to explain this, at least in the supplementary materials.

- There are many typos in the manuscript. The authors should carefully review the grammar if the paper was accepted.


**Questions:**

- The paper says that $T(\lambda) \sim \mathcal{N}(0,1)$, but looking at Figure 4, the scores do not appear to be Gaussian, and the scale of the scores grows to several thousands. Is my understanding about Fig. 4 wrong, or there is some mistake in plotting the figure?

- To computer $RHT(\lambda)$ in equation (3), one needs to invert the matrix $S_n + \lambda I_p$, which has size $p$ that is the dimension of the latent representation. What if $p$ grows to, say one million? Would it still be possible to compute the score? Also the paper does not seem to clearly state the runtime of the proposed algorithm.

---

> ### Author Rebuttal · Authors · 2023-08-09
>
> We greatly appreciate your positive and insightful comments on our manuscript.  We would address your comments as follows.
>
> >**Q1. More Explanation on ARHT.**
>
> We understand the audience without a solid statistical background may find it difficult to understand the proposed ARHT. We provide more explanations of ARHT as follows and will revise the manuscript for a clearer explanation of key statistical components to the general audience.
>
> *  **Why ARHT?** One of our motivations is to formulate uncertainty estimation as a hypothesis testing problem, where we interpret high-dimensional test statistics as **distance measures**. ARHT can be viewed as a distance measure between each testing sample and the in-distribution sample, where a larger ARHT indicates the sample is more likely to be OOD. The detection of OOD is then determined by a threshold (we presented the Benjamini--Hochberg (BH) procedure to compute the optimal threshold). One advantage of ARHT is that it directly operates on the sampling distributions of the latent features, and hence it does not require parametric assumptions on the latent features or logits (e.g., Dirichlet distribution on class probabilities), which makes it a more robust metric for uncertainty estimation.
>
> * **Motivation of Loading $\lambda I$.** This is a standard way to ensure the covariance matrix is positive definite (and thus invertible) and hence improve the numerical stability. We will state this earlier in the article to avoid confusion.
>
> * **Why ARHT follows standard normal?**  The major part of Li *et al.* is to prove why $T(\lambda)$ follows the standard Gaussian distribution. The proof is complex and hence is not focused in this paper. Intuitively, $T(\lambda)$ can be viewed as a ``standardized" version of RHT (i.e., known as Mahalanobis distance) by its theoretical mean and SD (where the derivations of the theoretical mean and SD are also specified in detail by Li *et al.*
>
> * **Intuitive Explanation and detailed derivation of the BH procedure.**
>     We agree that the BH procedure may not be intuitive to the general audience although it is well-known in the statistics community (due to multiple testing). Intuitively, we consider the OOD detection procedure for each testing image as a single hypothesis testing problem. Then, such a procedure for the whole testing set can be viewed as a multiple-testing problem (i.g., conducting many tests). However, applying a universal threshold for all tests (e.g., $\alpha =0.05$) is too conservative and leads to many false discoveries. Hence, the BH procedure is applied to assign a threshold adaptively to each sample according to the $p$-values of all tests such that the false discovery rate (FDR) can be minimized.
>
> >**Q2. Plot of $T(\lambda)$.**
>
> Yes, this observation is inconsistent with the theoretical property of $T(\lambda)$. In fact, this plot is very similar to the $F$-distribution of Mahalanobis distance. We believe this should result from a partial violation of the assumption of ARHT (most likely the covariance scale is heterogeneous (i.e., unequal variance testing) between the training and testing distributions). However, other empirical evaluation shows that the violation of assumption is not significant on performance, and we believe this violation only affect the scale of the test statistic. We will further develop a variance-adjusted version of ARHT as an attempt to resolve this issue.
>
> >**Q3. Dimension of $p$.**
>
> Since we are focusing on testing on latent representation, it is uncommon that the dimension of latent representation is too large (say one million). Typical dimensions of embeddings range from 64 to 1024, and we have conducted ablation studies on how it affects performance. Indeed, the costs of inversion increase significantly with the increase in dimension. Therefore, we rely on the assumption that the encoder can generate good feature distributions such that our method is less sensitive to the feature dimension (i.e., a feature dimension of 128 can be sufficient). The runtime of the algorithm is majorly on computing the inverse of the sampled covariance matrix and the multiple forwards of the Bayesian Neural Network (i.e., getting distributions of latent representation). The inference runtime is around 20 seconds per batch (size: 1024) of samples when $n_2=200$ and $p=64$.
>
> >**Q4. Typos in the Manuscript.**
>
> Thank you for pointing out the issues. We have conducted a thorough review of the typos and grammar and will update the typo-free manuscript at a later stage.

---

> > ### Comment · Reviewer_VpVz · 2023-08-18
> >
> > Thanks for the response, I maintain my original score.

---

### Official Review · Reviewer_jk7P · 2023-07-02

**Soundness:** 3 good
**Presentation:** 3 good
**Contribution:** 3 good
**Rating:** 6
**Confidence:** 4

**Summary:**

The paper proposes BNN-ARHT, which introduces a uncertainty estimation framework that uses high dimensional hypothesis testing in the feature space of a network. The key idea is to use ARHT to determine in vs outliers in a feature space, and so it is generic and broadly applicable to any kind of task and network type trained with supervised or unsupervised objectives. In particular, the paper relies on a Bayesian encoder to compute the ARHT on train and test data, that is trained with Stochastic Variational Inference (SVI). The testing statistic is the Hotelling’s $T^2$ test statistic, of which the adaptable and regularized version is used. The threshold $\lambda$ is tuned from a pre-defined set for each sample, and for each task separately which is argued to produce the best results

**Strengths:**

* ARHT seems to be a interesting and novel statistic from the uncertainty estimation point of view.
* An appealing aspect of the proposed approach is that it can work in arbitrary feature spaces, which vastly expands its applicability. This is also relatively under studied as compared to most approaches that compute uncertainty in the output space.
* Predictive performance on the networks tested is preserved, while retaining benefits of OOD detection. Good ablation studies are performed to understand the characteristics of the method.
* The finding that the method can be used with a encoder trained with supervised or unsupervised architecture is very interesting, and raises several intriguing questions. This can be studied in more detail — perhaps a related work is "Predictive Inference with Feature Conformal Prediction” from ICLR 2023. https://arxiv.org/abs/2210.00173

**Weaknesses:**

* The paper only compares with few uncertainty estimation methods for the OOD detection setup — whereas i think it should compare more comprehensively with the wide range of state of the art techniques that have achieved high performance on all the benchmarks considered here.
* The lack of more realistic OOD benchmarks is a limitation. The experiments are only done on CIFAR10/MNIST with simple networks (ResNet18) whereas it is essential to validate these interesting claims on more realistic benchmarks.
* The generality of the approach needs to be rigorously evaluated — UQ for regression, and tasks other than classification can be evaluated and compared against existing methods.
* On UQ -- why is OOD the only application considered? OOD benchmarks are often artificial and can have a lot of artifacts which can aid in detection (See for e.g. Semantically Coherent Out-of-Distribution Detection, ICCV '21). Tasks like Bayesian function optimization can shed more light on the quality of uncertainties, and its performance on regression.
* Scalability needs to be addressed — a big limitation is the scalability of the proposed approach to bigger and more realistic datasets and networks. Even for ResNet18 considered here, the only a single layer is replaced with its BNN equivalent. it’s unclear why the last layer is optimal, or how this choice can be made in other settings. Further, the threshold needs to be identified each time for each task, which is a computational burden, that other competing methods do not have.

**Questions:**

My main concerns are regarding the lack of more comprehensive evaluation for OOD detection. Using better baselines, large scale benchmarks and more modern architectures.

**Limitations:**

Yes

---

> ### Author Rebuttal · Authors · 2023-08-09
>
> Thank you for your positive comments on the novelty and the appealing aspects of our methods. And thank you for indicating an intriguing future direction for this work using conformal inference. We will address your concerns and questions as follows.
>
> >**Q1. More Comprehensive Evaluation.**
>
> According to your comments, we conduct additional experiments to more extensively evaluate our method, including adding more baselines, experiments on more diverse datasets, and experiments with larger architectures.
> * **More baselines.** We have added additional baselines on uncertainty estimation approaches for comparison: (1) I-EDL [1]: use the Fisher information matrix to measure the informativeness of evidence carried by each sample; and (2) RKL-PN [2]: prior networks trained with the reverse KL divergence. **Table 1 in the rebuttal file** presents the results. We will comprehensively compare these baselines on more datasets in the future.
>
> *  **More datasets.** We validated our framework on a more realistic dataset in medical imaging --- the Diabetes Retinopathy Detection (DRD) dataset (see Table 2 in the manuscript for the results) in addition to standard datasets like CIFAR and MNIST. Additionally, we performed uncertainty estimation on a larger and more diverse dataset (as suggested by reviewer vwXa) with TinyImangeNet as the OOD dataset. The results show that our model is still robust when generalized to a larger dataset. Please see the **general comments** for the results and a detailed discussion.
>
> *  **More tasks.** We choose OOD as the benchmark task since it is the most common benchmark for uncertainty estimation. We additionally construct a regression setting with two multivariate Gaussian distributions with different means and variances indicating different distributions. Most of the uncertainty estimation frameworks cannot apply in this setting since they only work under the classification settings. The following table presents the results. The Experiment demonstrates that our method also achieves satisfactory performance under the **regressions settings**, showing the generalizability to other tasks.
>
> Table: The OOD detection performance (in \%) of our method compared with various competitors, using the ResNet50 architecture. We constructed a simulated regression setting where $\mathcal{N} (\boldsymbol \mu, \boldsymbol \Sigma)$ is the distribution for in-distribution data and $\mathcal{N} (-\boldsymbol \mu, \boldsymbol \Sigma)$ is the distribution for OOD data. The auxiliary regression task is to predict the norm of the sampled vector with an MLP with 2 layers. We choose $\boldsymbol \mu = [0.5, 0.5, \ldots, 0.5]^\top$ and $\boldsymbol \Sigma = 9 I$.
> |  Model   | AUROC  | AUPR |
> |  ----  | ----  | ---- |
> | MC Dropuout | 62.12  | 63.35
> | Deep Emnsembles | 73.18 | 70.45
> |Kendall and Gal | 67.00 | 70.00
> | BNN-ARHT (Ours)  | **73.52** | **72.99**
>
> >**Q2. More Modern Architecture \& Scalability concerns.**
>
> Following your suggestion, we additionally tested the performance of our ARHT using ResNet 50 with some layers being replaced by its Bayesian replication. The results show that our method still performs well when being scaled to large model architecture. Note that our method does not necessarily rely on a Bayesian encoder. In fact, a large frequentist network (e.g., ViT) can also generate ideal latent distributions, while our experiments focus on the BNN to emphasize its capability to generate good latent distributions. Detailed results and discussion can be found in the **general comments**.
>
> >**Q3. Replacing Some Layers to Create Deep Bayesian CNNs.**
>
> The rationale of replacing some layers (either convolutional or linear layer) as Bayesian is to introduce variance to parameters. The empirical performance demonstrates that our method is **less sensitive to which layers are being replaced** (we experimented on several combinations of layers), as long as the total variance in parameters is not too large (such that the BNN would not underfit to a random guess).
>
> >**Q4. Identification of Threshold $\lambda$.**
>
> We consider the threshold $\lambda$ as a characteristic in our algorithm which makes the uncertainty estimation adaptive.  We understand the concern about the computational burden.  However, since the computation of the inverse of the covariance matrix dominates the algorithm, the computation of the optimal $\lambda$ is relatively trivial.  Moreover, from the ablation study, we observe that the framework is robust to changes in $\lambda$, and thus one can also set a universal $\lambda$ in practice.
>
> >**Q5. Uncertainties with Conformal $p$-Values.**
>
> We would like to thank the reviewer for highlighting an interesting direction of uncertainty estimation with conformal inference. Currently, our settings assume the samples drawn from the latent distribution are IID (also the assumption of ARHT). While this may be true for the in-distribution data, it is likely that the IID assumption is violated for testing samples (as latent features are drawn from a single testing image, which leads to highly correlated features). This is also a possible reason why Figure 4 is not strictly the standard normal (see the discussion with reviewer VpvZ). With conformal inference relaxing the IID assumption to exchangeability, we may be able to develop more robust P-values as uncertainty estimation measures in future work (a related work is Testing for Outliers with Conformal $p$-values S Bates *et al.* 2021)
>
> [1] Deng *et al.* Uncertainty estimation by fisher information-based evidential deep learning. ICML 2023
>
> [2] Malinin and Gales. Reverse kl-divergence training of prior networks: Improved uncertainty and adversarial robustness. NeurIPS 2019

---

> > ### Comment · Reviewer_jk7P · 2023-08-14
> > **response**
> >
> > Dear authors, thank you for the rebuttal. This answers some of my questions, but many still remain. First, i still think scaling is a big weakness of the proposed approach. While it is encouraging to see it being scaled to Resnet-50, i find the performance quite surprising. Weaker models on Cifar-10/SVHN have performed far superior to the numbers reported here -- can the authors explain why? For e.g. Deep Ensembles has an AUROC of only 65 here, where as previously published work has shown even with Resnet-18 to have AUROC of 90+ even with a 3-model ensemble. (See table 3, http://proceedings.mlr.press/v119/van-amersfoort20a/van-amersfoort20a.pdf). Is this because of the scoring function used?
> >
> > Even a simple MSP with resnet-18 should give high AUROC on CIFAR-10/SVHN, which makes the justification of the proposed approach weaker.
> >
> > Next, by scalability of datasets, I meant the in distribution datasets (such as ImageNet) and not necessarily OOD sets like TinyImageNet -- though this is still good evidence to have in support of your technique.
> >
> > Next, by regression tasks, I was primarily talking about uncertainty in regression settings which is much broader (such as function optimization) but it is encouraging to see the synthetic experiment nevertheless.

---

> > > ### Author Response · Authors · 2023-08-17
> > >
> > > Thank you for your prompt reply. We will address your concerns as follows.
> > >
> > > > **Q1 Better Performance observed for Weaker Models.**
> > >
> > > We are also aware of this phenomenon in experiments. The reason may be several points. First, the CIFAR 10 is a relatively small dataset (i.e., the training set is small compared to its complexity, also mentioned in [2]), so it is not easy to optimize large model architectures with such a small dataset. Second, a deeper architecture may not always generate good latent distribution. Although the mean generated is close to the true mean (with a deeper architecture), the covariance structure of the embedding may not be well-approximated. Hence, the ARHT may obtain lower OOD detection performance even with a very good frequentist encoder. We have discussed this (i.e., the difference between frequentist and BNN encoder) in detail in the discussion section of the manuscript.
> > >
> > > Moreover, we perform additional experiments using TinyImageNet as the training/in-distribution dataset and CIFAR10 as the OOD dataset to show the case where a larger architecture takes advantage. We observe that a larger architecture (i.e., ResNet50) obtains better performance in this case. Furthermore, since ARHT is determined by many factors, one may not always observe a monotonously improving performance with larger architectures. We have conducted a thorough **ablation study in the manuscript and in the discussion with reviewer HkqB** on the material factors affecting the performance. We will perform a more thorough analysis of the effect of deeper architectures in future works beyond the analysis in **Table 1**.
> > >
> > > Table 1 The OOD detection performance (in \%) of our method using different architectures. We use TinyImageNet as the in-distribution dataset and CIFAR10 as the OOD dataset.
> > > |  Model   | AUROC  | AUPR |
> > > |  ----  | ----  | ---- |
> > > | LeNet | 69.27|71.54
> > > | ResNet50 | 71.50| 72.68
> > >
> > > > **Q2 Performance of ResNet-based Deep Ensemble.**
> > >
> > > Thank you for introducing the related work. For the implementation of deep ensembles, we adopt the original implementation and its scoring function [1], while [2] re-implemented the scoring function with their own design. We think there are several reasons for this phenomenon. Firstly, [2] improves the classic uncertainty measure for classification (i.e., entropy) as the distance to the closest latent centroid. This addresses the epistemic uncertainty in the model. Secondly, they also regularize the learning process with gradient penalty so that the model (especially the large model) is less easy to overfit, making the ResNets in the ensemble more well-trained. Thirdly, the proposed method in [2] is also limited to the classification problem only so it is expected that the uncertainty estimation performance under the classification settings is better.  These factors greatly improve the original implementation of the deep ensemble on OOD detection performance. Hence, it is reasonable that the refined deep ensemble obtains a high performance.
> > >
> > > [1] Lakshminarayanan et al. Simple and Scalable Predictive Uncertainty
> > > Estimation using Deep Ensembles. NeurIPS 2017.
> > >
> > > [2] Amersfoort et al. Uncertainty Estimation Using a Single Deep Deterministic Neural Network. ICML 2020.
> > >
> > > > **Q3 ImageNet as in-distribution dataset.**
> > >
> > > Thank you for your clarification. Per your suggestion, we conducted additional experiments with ImageNet as the in-distribution dataset and CIFAR10 as the OOD dataset. Please see the table below for the results. The results validate that our method can also generate well when the feature encoder is trained at a more diverse and complex dataset (i.e., TinyImageNet).
> > >
> > > |  Model   | AUROC  | AUPR |
> > > |  ----  | ----  | ---- |
> > > | MC Dropuout | 64.36| 60.47
> > > | Deep Ensembles | 66.41| 63.97
> > > |Kendall and Gal | 58.54| 55.29
> > > |EDL |50.37| 70.39
> > > |DPN|59.59| 59.87
> > > | BNN-ARHT (Ours)  | **69.27** | **71.54**
> > >
> > > > **Q4 Generalization of Tasks.**
> > >
> > > Thank you for your appreciation of our simulation experiments and for suggesting Bayesian optimization as a potential application of our work. We are aware that Bayesian optimization is a well-known field of optimization. However, the definition of optimizer uncertainty is slightly different from ours. To the best of our knowledge, Bayesian optimization is concerned with the uncertainty from **optimizer** instead of the uncertainty from data and model [3]. The model quantifies the uncertainty as the **posterior distribution** given the trajectory at step $t$ (please see [3] for a formal definition), which we think can be better approximated by a BNN instead of a distance measure (e.g., ARHT or Mahalanobis distance). Extension of our method to Bayesian optimization is possible, but non-trivial, which is not the focus of our paper, although we agree this is an interesting and exciting direction to work on in the future.
> > >
> > > [3] You et al. Bayesian Modeling and Uncertainty Quantification for Learning to Optimize: What, Why and How. ICLR 2022

---

> > > > ### Comment · Reviewer_jk7P · 2023-08-18
> > > >
> > > > Thanks for the response and clarifications. I am happy with the clarification and will increase my score.

---

### Official Review · Reviewer_vwXa · 2023-07-10

**Soundness:** 3 good
**Presentation:** 3 good
**Contribution:** 2 fair
**Rating:** 6
**Confidence:** 3

**Summary:**

This paper proposes an OOD detection procedure by applying adaptable regularized Hotelling’s T-square (ARHT) test [24] on the feature representation of learned BNN networks. Authors introduced the application o ARHT on BNN encoder, and proposed a procedure to adaptively calibrate detection threshold based on Benjamini–Hochberg (BH) procedure. On a suite of low-dimensional image experiments (CIFAR, MNIST, OMNIGLOT, SVHN) and a medical image benchmark (DRD) and using small architectures (e.g., LeNet, AlexNet, ResNet18), authors illustrates that the proposed method clearly outperforms previous methods in terms of OOD AUC. Thorough ablation study is conducted.

**Strengths:**

* An novel approach to OOD detection leveraging multivariate tests and associated calibration procedures.
* Authors conducted thorough experiment on academic benchmarks to show the advantage of the approach.

**Weaknesses:**

* Some description of previous literature may be incorrect. For example, in line 43-48, authors mentions that (1) BNNs perform poorly when the dimension of the output is high. (2) BNN are limited to classification problems. I am not sure if either of these are true for modern BNNs. For example, [rank-1 BNN](https://arxiv.org/pdf/2005.07186.pdf) outperforms their deterministic counterparts on ImageNet, and regression is a rather standard BNN task (e.g., conducted on UCI benchmarks).

* The experiments are done on rather simplistic architectures and benchmarks.  Therefore the generalization of this approach (either in terms of quality and computational feasibility) to more realistic data setting (e.g., ImageNet) and nontrival modern large models is less clear.

* I find that the point "our framework is sensitive to the quality of the encoder" (line 260-266) to be rather important. Three suggestions regarding this point:

   * this seems to contradict the paper's claim that ARHT is superior to traditional BNN whose "performances heavily rely on the feature encoder and are poor when the features are of poor quality" (line 48). As it seems ARHT suffer a similar issue but to a lesser extent. I recommend adjust the descriptions in 48-50 correspondingly so it is not misleading (maybe refocus (3) in terms of sample efficiency).
  * It would be interesting to have a more thorough investigation on the relationship between encoder quality v.s. OOD detection performance, where encoder quality can be quantified in terms of standard representation learning metric such as linear probing accuracy.  This can be done via a scaling study of different sample sizes and / or architectures. I think a study like this can possibly show ARHT performance is associated with encoder quality, but can achieve a stronger OOD performance when compared to other method under the same accuracy.
  * Please consider discussing this in the limitation section.

**Questions:**

See Weakness.

**Limitations:**

See Weakness.

---

> ### Author Rebuttal · Authors · 2023-08-09
>
> Thank you for your detailed and constructive comments. We will address your questions and concerns as follows.
>
> >**Q1. A More accurate description of the literature.**
>
> Thank you for the suggestions. We believe that there might be some misunderstanding. We focused on comparing the previous uncertainty estimation methods instead of BNNs. Most of the **uncertainty estimation methods** (with or without BNNs) are limited to classification problems. They do not operate on latent distributions, hence they perform poorly when the dimension of output is high (e.g., high dimensional regression). Hence, we highlight that our proposed uncertainty estimation method can address these common limitations. We will revise the description in our manuscript for a clearer presentation to avoid confusion.
>
> >**Q2. Generalization to More Realistic Datasets and Modern Architectures.**
>
> We have included additional experiments on TinyImageNet and ResNet50. The results still outperform the baseline methods, validating the performance of our framework when being scaled to larger datasets and architecture.  Please see the **general comments** for the results on TinyImageNet and the **comments to reviewer jk7P** for results with ResNet50.
>
> >**Q3. Influence of the Quality of Encoder.**
>
> Thank you for the suggestions. We are also aware of the effect of the encoder quality, and thus we fixed the encoder (say, LeNet) when comparing our framework with the current SOTA methods. Since the baseline methods are trained on classification problems (e.g., the CIFAR 10 image classification and DRD auxiliary task), we adopt the **training accuracy at different epochs** to measure the quality of the encoder, and we observe that the OOD detection performance is overall monotonously improved with the increase in training accuracy (i.e., the encoder quality). See **Figure 3 in the rebuttal file** for the visualization. More specifically, we claim that ARHT outperforms traditional **uncertainty estimation methods** which heavily rely on the encoder quality. Since ARHT operates on latent representations, it still requires the quality of the encoder to generate ideal latent distributions. We would adjust the claim in a later version of the manuscript and clarify this point in the limitation section.

---

### Author Rebuttal · Authors · 2023-08-09

We thank all the reviewers for your time and efforts on our manuscript. According to the reviewers' comments, we conduct additional experiments to more extensively evaluate our method, including adding more baselines, experiments on more diverse datasets, and experiments with larger architectures. Experiment results are summarized in the rebuttal file and tables in respective threads.

>**Q1. More realistic datasets (jk7P, vwXa).**

We have conducted additional experiments on OOD detection, with CIFAR10 as the in-distribution dataset and TinyImageNet as the OOD dataset. The result is presented in Table 1 below and it shows that the uncertainty estimation performance is still satisfactory when being generalized to larger datasets.

Table 1: The OOD detection performance (in \%) of our method compared to various competitors, using the LeNet architecture. We use CIFAR 10 as the in-distribution dataset and TinyImageNet as the OOD dataset.
|  Model   | AUROC  | AUPR |
|  ----  | ----  | ---- |
| MC Dropuout | 66.98  | 64.46
| Deep Emnsembles | 66.41 | 63.97
|Kendall and Gal | 63.23 | 63.06
|EDL | 51.64 | 66.31
|DPN| 64.68 | 58.33
| BNN-ARHT (Ours)  | **67.77** | **66.74**

>**Q2. Scalability (jk7P, vwXa, HkqB).**

 We aim to use the Bayesian counterpart of a smaller architecture to demonstrate the capability of BNNs to generate latent feature distributions (for details please see the discussion section in the main text). Note that one can generate ideal feature distributions using very large frequentist vision models (e.g., ViT), which however induces larger complexity in training and inference.

A related ablation experiment comparing the frequentist and Bayesian architecture is presented in the main text. We have additionally conducted an experiment with the Bayesian model architecture scaled up to ResNet50. Table 2 below presents the results and it shows that our method can still perform satisfactorily when scaled up to a larger architecture. We further conduct an example experiment using the frequentist ResNet50 (the hypothesis test reduces to a one-sample test) and the testing AUROC is 72.77. These results validate the scalability of our method to large and modern vision architectures.

Table 2: The OOD detection performance (in \%) of our method compared with various competitors, using the ResNet50 architecture. We use CIFAR 10 as the in-distribution dataset and SVHN as the OOD dataset.
|  Model   | AUROC  | AUPR |
|  ----  | ----  | ---- |
| MC Dropuout | 68.32  | 78.24
| Deep Emnsembles | 65.13 | **82.19**
|Kendall and Gal | 72.24 | 81.43
|EDL | 51.21 | 73.78
|DPN| 62.33 | 79.11
|Detectron | 73.16 | 82.5
| BNN-ARHT (Ours)  | **73.46** | 78.27

---

### Decision · Program_Chairs · 2023-09-21

**Decision:**

Accept (poster)

**Comment:**

This paper proposes a new method for uncertainty estimation by (1) mapping data into a latent feature space using Bayesian Neural Networks and (2) run an adaptable regularized Hotelling’s $T^2$ (ARHT) test for better consistency and robustness especially for relatively high latent dimension. The method was tested on OOD tasks and benefits were shown.

The reviewers are generally in favor of the paper. They considered the idea novel and effective. Different concerns such as scalability, generalizability, experiment setting, etc. were raised but were addressed reasonably during rebuttal.